# Footprint of the host restriction factors APOBEC3 on the genome of human viruses

**Florian Poulain**\*, **Noémie Lejeune, Kévin Willemart, Nicolas A. Gillet**⊙\*

Namur Research Institute for Life Sciences (NARILIS), Integrated Veterinary Research Unit (URVI), University of Namur, Namur, Belgium

\* poulainflorian@gmail.com (FP); nicolas.gillet@unamur.be (NAG)

**Data Availability Statement:** S1 Table, S2 Table and S3 Table files are available from the Dryad database via the link: https://datadryad.org/stash/

## Abstract

APOBEC3 enzymes are innate immune effectors that introduce mutations into viral genomes. These enzymes are cytidine deaminases which transform cytosine into uracil. They preferentially mutate cytidine preceded by thymidine making the 5'TC motif their favored target. Viruses have evolved different strategies to evade APOBEC3 restriction. Certain viruses actively encode viral proteins antagonizing the APOBEC3s, others passively face the APOBEC3 selection pressure thanks to a depleted genome for APOBEC3-targeted motifs. Hence, the APOBEC3s left on the genome of certain viruses an evolutionary footprint.

The aim of our study is the identification of these viruses having a genome shaped by the APOBEC3s. We analyzed the genome of 33,400 human viruses for the depletion of APOBEC3-favored motifs. We demonstrate that the APOBEC3 selection pressure impacts at least 22% of all currently annotated human viral species. The *papillomaviridae* and *polyomaviridae* are the most intensively footprinted families; evidencing a selection pressure acting genome-wide and on both strands. Members of the *parvoviridae* family are differentially targeted in term of both magnitude and localization of the footprint. Interestingly, a massive APOBEC3 footprint is present on both strands of the B19 erythroparvovirus; making this viral genome one of the most cleaned sequences for APOBEC3-favored motifs. We also identified the endemic *coronaviridae* as significantly footprinted. Interestingly, no such footprint has been detected on the zoonotic MERS-CoV, SARS-CoV-1 and SARS-CoV-2 coronaviruses. In addition to viruses that are footprinted genome-wide, certain viruses are footprinted only on very short sections of their genome. That is the case for the *gamma-herpesviridae* and *adenoviridae* where the footprint is localized on the lytic origins of replication. A mild footprint can also be detected on the negative strand of the reverse transcribing HIV-1, HIV-2, HTLV-1 and HBV viruses.

Together, our data illustrate the extent of the APOBEC3 selection pressure on the human viruses and identify new putatively APOBEC3-targeted viruses.

## Author summary

APOBEC3 cytidine deaminases are enzymes that restrict many viruses by mutating their genomes. In doing so, they exert a selection pressure and leave onto these viruses an

share/VikUp3LVVUInFa88SbRwP4JGQu
PpehlvcqdZSBevHwY.

**Funding:** This study was supported by F.R.S.-
FNRS (Fonds de la Recherche Scientifique de
Belgique, https://www.frs-fnrs.be) grant CDR No.
31270116 and by the University of Namur. F.P. is a
PhD fellow supported by a NARILIS (https://www.
narilis.be) research grant and by F.R.S.-FNRS
Télévie (https://www.frs-fnrs.be) grant PDR-TLV
No. 34972507. N.L. is a PhD fellow supported by
FRIA (Fonds pour la Formation à la Recherche
dans l'Industrie et dans l'Agriculture, https://www.
frs-fnrs.be) grant No. 31454280. Computational
resources have been provided by the Consortium
des Équipements de Calcul Intensif (CÉCI), funded
by the F.R.S.-FNRS under grant No. 2.5020.11 and
by the Walloon Region (https://recherche-
technologie.wallonie.be). The funders had no role
in study design, data collection and analysis,
decision to publish, or preparation of the
manuscript.

**Competing interests:** The authors have declared
that no competing interests exist.

evolutionary footprint. In addition to their antiviral role, APOBEC3s have also been iden-
tified as a major source of mutations in cancer, wrongly targeting the cell genome. For
example, high-risk papillomaviruses, whose viral genomes carry an APOBEC3 footprint,
indirectly promote cell transformation due to the sustained APOBEC3 mutagenic activity.
In this study, we perform for the first time a general screening for the APOBEC3 footprint
in all currently annotated human viruses. We show that approximately 22% of human
viral species are shaped by the APOBEC3 selection pressure and extend the list of APO-
BEC3-footprinted viruses with adenoviruses and autonomous parvoviruses. Knowing
which virus is restricted by the APOBEC3 mutagenic activity could lead to the identifica-
tion of new viruses associated with cancer.

## Introduction

The APOBEC3s (apolipoprotein B mRNA-editing enzyme, catalytic subunit 3 or A3s) are
innate immune effectors restricting many exogenous viruses and endogenous retroelements
[1–3]. The human genome encodes for seven A3 genes (namely A3A, B, C, D, F, G and H),
with several spliced transcripts and allelic variants for each. These seven genes originate from
gene duplications and rearrangements that have occurred during mammalian evolution and
represent a classic example of the virus-host arms race [4]. The A3s are cytidine deaminases
that convert cytosine to uracil present in single stranded DNA or RNA. Such editing on viral
genomes generally results in C to T (or U) transition after replication of the genome. The A3s
preferably mutate cytosine in a 5'TC dinucleotide context with the notable exception of A3G
that favors a C before the mutated C [5].

The antiviral activity of the A3s has been first reported for the reverse transcribing viruses
HIV-1 (human immunodeficiency virus-1), HTLV-1 (human T-lymphotropic virus-1) and
HBV (Hepatitis B virus) [6–8]. Editing occurs during reverse transcription on the negative
strand leading to G to A mutations on the positive strand [9–14]. Subsequently, A3-introduced
mutations have been reported on various double-stranded DNA (EBV for Epstein-Barr virus,
HSV-1 for herpes simplex virus-1, α-HPVs for alpha human papillomaviruses, BK PyV for BK
polyomavirus), single-stranded DNA (TT virus) and single-stranded RNA (HCoV-NL63 for
human coronavirus NL63) viruses [15–19]. It is important to note that the antiviral action of
the A3 proteins is not based solely on their deaminase activity. Deaminase-independent
restriction has been demonstrated against endogenous retroelements, reverse transcribing
viruses, adeno-associated viruses and many RNA viruses (HCV for hepatitis C virus, RSV for
respiratory syncytial virus, HCoV-NL63, mumps virus and measles virus) [20–25].

The co-evolution between virus and host leads to the selection of viral proteins capable of
countering the restriction effect of A3s. HIV-1 encodes for the Vif protein which promotes
A3G degradation [26]. HTLV-1 evades A3G restriction by excluding A3G from virions [27].
BORF2 protein from EBV inhibits A3B deaminase activity and re-locates it far from viral repli-
cation centers [28]. Besides these active viral mechanisms targeting A3 proteins, some viruses
have evolved passive strategies to limit A3 restriction. One such strategy is the depletion of
A3-favored motifs from the viral genome. By repetitive exposure to A3 activity, non-lethal
mutations can accumulate in the genomic sequence leading to the under-representation of the
motifs favored by A3s. This under-representation of A3-favored motifs is called A3 evolution-
ary footprint. Thus, the 5'TC dinucleotide motif is under-represented in the genome of α-
HPVs and BK polyomavirus [17,29]. Similarly, 5'TC and 5'CC (favored by A3G) motifs are
under-represented in the negative strand of LTR (long terminal repeat) and non-LTR

endogenous retroelements [30]. Conflicting data have been reported regarding evidence of an A3 footprint on the HIV-1 genome [31,32]. Recently, an under-representation of certain A3 motifs has been shown in the genome of the γ-herpesviruses EBV and KSHV (Kaposi sarcoma herpes virus) [33]. Finally, codon usage in coronaviruses suggests that cytosine deamination is an important biochemical force which shapes the evolution of these viruses [34].

Different bioinformatics approaches have been used to search for evidence of an A3 evolutionary footprint in viral genomes [17,29–33,35,36]. In this study, we adapted and extended the Warren *et al.* approach [29] to carry out a general screening for the A3 footprint of the genomes of all currently annotated human viruses. We first demonstrate the sensitivity and specificity of our approach: i. an A3 footprint is detected in viruses which have already been shown to be depleted for A3-targeted motifs; ii. no A3 footprint has been reported in viruses from animals lacking A3 genes. We showed that as much as 22% of currently annotated human viral species are shaped by the A3 selection pressure. We confirmed previous reports showing that papillomaviruses and polyomaviruses are generally strongly footprinted by the A3s. Among the most A3-footprinted viruses, we identified autonomous parvoviruses and in particular the B19 erythroparvovirus as deeply cleansed for A3-favored motifs. Importantly, we showed that the A3 footprint observed in coronaviruses is limited to the endemic viruses and absent from the zoonotic MERS-CoV (middle east respiratory syndrome coronavirus), SARS-CoV-1 (severe acute respiratory syndrome coronavirus) and SARS-CoV-2 viruses. We also carried out a gene-specific A3 footprint search and identified local footprint on EBV and adenoviruses consistent with genome targeting during the initiation of replication.

## Results

### Definition of the A3 footprint

The A3 footprint is defined as the under-representation of A3-targeted motifs. Different approaches have been devised to estimate the over/under-representation of a given motif [17,29–33,35,36].

Firstly, and because most of the A3 proteins (A3A, A3B, A3C, A3F and A3H) favors deamination of cytosine to uracil in a 5'TC dinucleotide context, we have chosen to look for the under-representation of the 5'TC motif. Moreover, as originally developed by Warren *et al.*, we refined our analysis by distinguishing the position of the 5'TC motif relative to the coding sequence. Namely, we differentiate three K-mers containing the TC motif; one K-mer having the C in the first position of the codon (NNTCNN), one K-mer having the C in the second position of the codon (TCN) and one K-mer having the C in the third position of the codon (NTC). A3-introduced deamination of cytosine in viral genome produces an uracil that can be fixed in the form of thymidine after genome replication. This transition will have different impacts depending on the position of the mutated C. The C to T mutation will be non-synonymous if the C is at the first or second position of the codon (Fig 1A, NNTCNN and TCN K-mers). However, if the mutated C occupies the third position of the codon, the C to T mutation will always be synonymous (Fig 1A, NTC K-mer). Therefore, A3-driven natural selection should deplete more intensively NTC codons than TCN or NNTCNN motifs (as in those cases the C to U mutation will impact the encoded amino acid). Obviously, A3 editing can also target the template strand where a C to T mutation will translate into G to A transition in the coding strand. Again, this transition will have different impacts depending on the position of the mutated G. The G to A mutation will be non-synonymous if the G is at the first or second position of the codon (Fig 1A, GAN and NGA K-mers). However, if the mutated G occupies the third position of the codon the mutation will be most of the time synonymous (Fig 1A, NNGANN K-mer). Because synonymous mutations are presumably more likely to be retained

**A.**

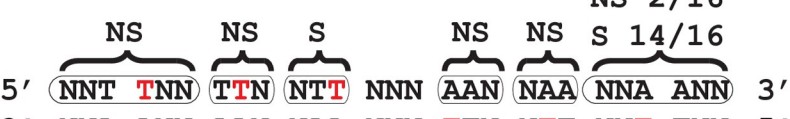

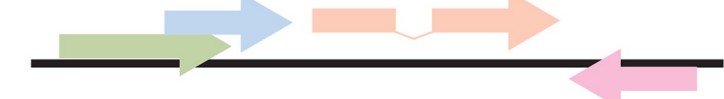

**B.**

**Viral genome**

**Concatemer of the viral coding sequences**

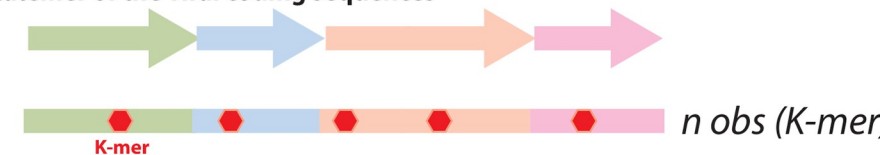

**Random shuffling of the concatemer**

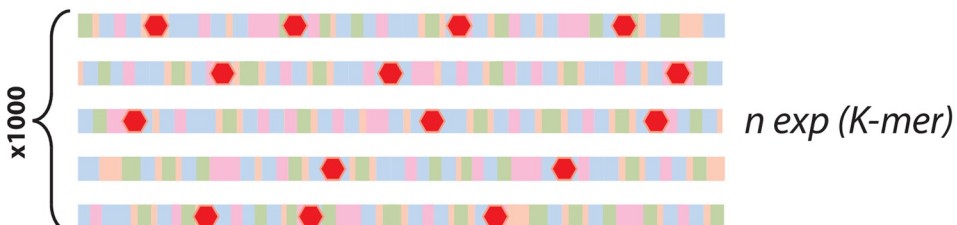

**NTC K-mer ratio calculation**

$$NTC\ ratio = \log_2 \left\{ \frac{n\ obs\ (NTC)}{n\ exp\ (NTC)} \right\}$$

**Fig 1. Definition and estimation method of the A3 footprint.** A. A3-induced cytidine deamination followed by viral replication leads to C to T mutations (in red). Most of the A3 enzymes favor deamination in a 5'TC context. The TC dinucleotide motif is depicted in three possible codon contexts on both coding and template strand. Depending on the position of the mutated C, the C to T transition can be synonymous (S) or non-synonymous (NS). Proportion of S and NS mutations is reported when the two types of mutation can be produced. Because synonymous mutations are more likely to be retained, the A3 footprint can be defined as the depletion of the NTC and/or NNGANN codons. B. Depletion or enrichment of a given K-mer (e.g. NTC) is calculated as the log2 ratio of the observed occurrence of that K-mer (n obs) divided by its expected occurrence (n exp). For each human virus, its coding sequences (colored arrows) are concatenated to generate a synthetic coding genome from which we obtain the n obs of a given K-mer. The synthetic coding genome is then shuffled a thousand times and the n exp is calculated as the average count for that K-mer.

than non-synonymous, we define the A3 footprint (with the exception of A3G-induced footprint) as the depletion of NTC or NNGANN K-mers. Calculation of observed vs expected K-mer ratio has been adapted from Warren *et al.* and detailed in the material and methods section. Briefly, a synthetic coding genome was generated by concatenating the different coding sequences allowing the counting of the occurrence of a given K-mer (Fig 1B, n obs (K-mer)). Each synthetic coding genome has been randomly shuffled a thousand times. The expected count is calculated as the average of the occurrences of this K-mer over the thousand iterations (Fig 1B, n exp (K-mer)). A negative K-mer ratio indicates depletion of that K-mer. The observed vs expected ratio of the NTC K-mer will be compared to those of the NNTCNN and TCN K-mers. Similarly, the observed vs expected ratio of the NNGANN K-mer will be compared to those of the GAN and NGA K-mers. Of note, for the sake of clarity and simplicity, we have chosen to stick with a DNA genetic code throughout the manuscript. The reader will read a T as a U in the context of RNA viruses.

Secondly, and because A3G favors deamination of cytidine when preceded by another cytidine, we have chosen to look for the under-representation of the 5'CC motif. Following the same rationale, A3G-footprinted viruses should display to a stronger depletion of NCC codons compared to CCN or NNCCNN motifs (or a depletion of NNGGNN motifs versus the GNN and NGG motifs) (S1A Fig). The NCC ratio will be compared to those of the NNCCNN and CCN K-mers. Similarly, the NNGGNN ratio will be compared to those of the GGN and NGG K-mers.

## An A3 footprint is detected in viruses known to induce A3 expression

Given that the BK polyomavirus has recently been demonstrated to induce A3B expression and that it was depleted in 5'TC motifs [17], we considered this virus as a positive control to validate our approach. Fig 2A shows a strong depletion of the NTC motif. On the contrary, the dinucleotide 5'TC in the context of the NNTCNN and TCN K-mers are neither over- nor under-represented. The significant differences between the NTC ratio and the TCN (or NNTCNN) ratios reveal that the frequency of the TC motif is dependent on its position within a codon. It suggests that NTC depletion can be tolerated because of the degeneration of the genetic code. Importantly, the absence of NNTCNN and TCN depletion infers that this virus is still vulnerable to A3 restriction because deamination of those cytidines would lead to changes in the amino acid sequence. Fig 2B highlights the fact that NTC depletion is genome-wide and can be observed in each gene. Similarly, the 5'GA motif is significantly less abundant in the NNGANN context than in the GAN or NGA codons. The 5'GA depletion is also genome-wide (Fig 2B). We read these observations as the consequence of an A3 activity acting on both coding and template strands.

Extending our analysis to other polyomaviruses shows that both JC polyomavirus and Merkel cell polyomavirus bear an A3 footprint; footprint that is also genome-wide and present on both strands (Fig 2C–2F). The magnitude of the A3 footprint appears lighter on the Merkel

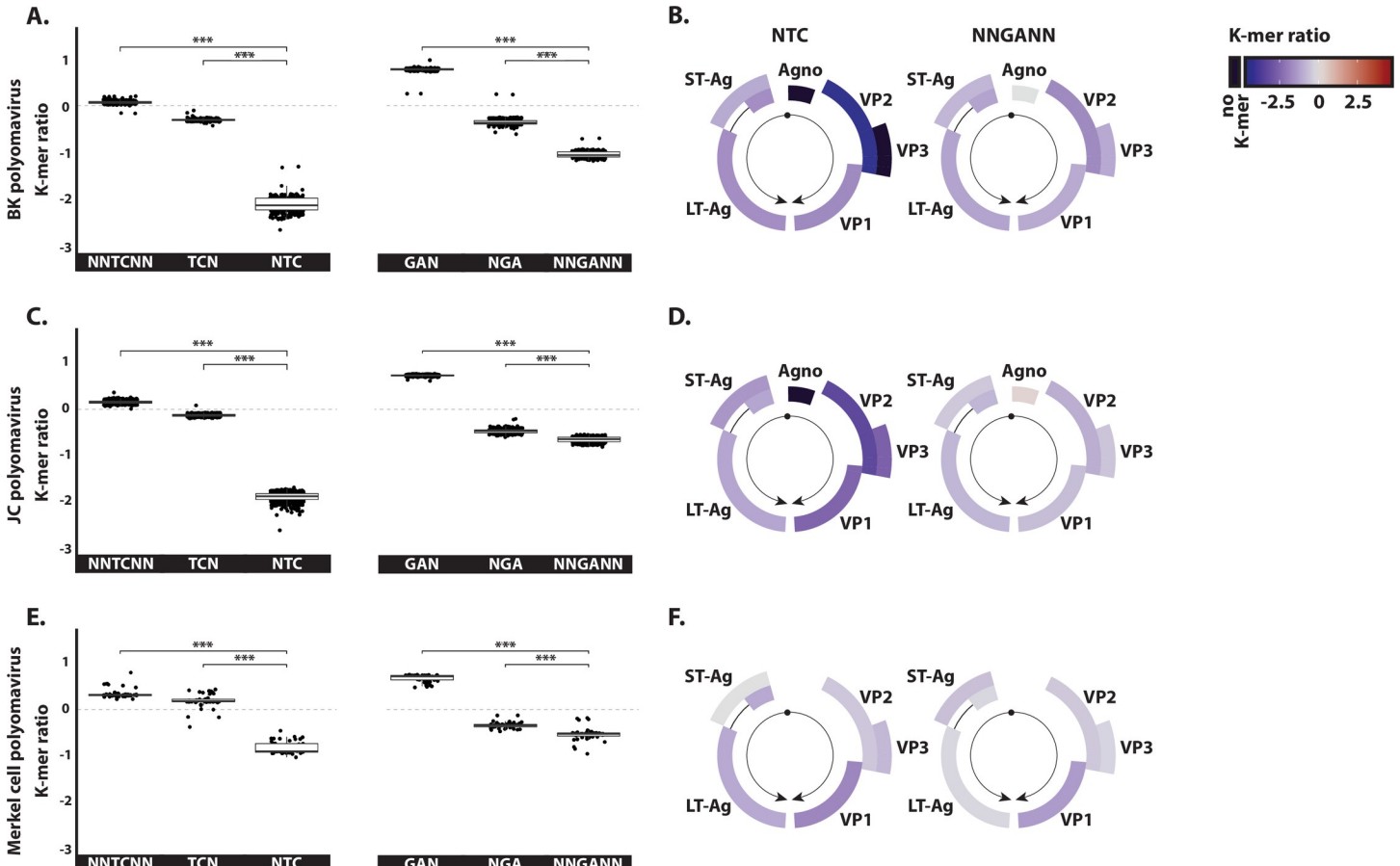

**Fig 2. Evidence of an A3 footprint in human polyomaviruses.** The observed/expected ratios of TC dinucleotide at various codon positions and on both strand (i.e. NNTCNN, TCN, NTC, GAN, NGA and NNGANN) were calculated for BK polyomavirus (panels A-B), JC polyomavirus (panels C-D) and Merkel cell polyomavirus (panel E-F). For the dot plots, each point stands for a unique full-length viral genome. Median and quartile are depicted by a boxplot. P-values were calculated by Student's unpaired, two-tailed t-test (NS for not significant, * p< 0.05, ** p< 0.01, *** p< 0.001). Panels B, D and F illustrate NTC and NNGANN ratios for the different viral coding sequences. A colored scale with increasing shades of blue indicating depletion and increasing shades of red indicating enrichment. Replication origin is illustrated by a black dot and gene transcriptional orientation is symbolized by black arrows.

cell polyomavirus. Analysis of a larger number of polyomavirus species shows a stronger NTC depletion in *beta-polyomaviridae* by comparison to *alpha-polyomaviridae* (S2 Fig). The *delta-polyomaviridae* are affected by an A3 footprint of variable intensity depending on the species considered (S2 Fig).

## The A3 footprint is limited to viruses infecting hosts endowed with A3 genes

We showed that our approach sensitively detects an NTC depletion in viruses known to promote A3 expression. We then wondered to what extent this depletion is widespread among viruses. We therefore downloaded genomic sequences of 33,400 human, 1,397 non-human primate, 9,160 avian, and 570 fish viruses and calculated NNTCNN, TCN and NTC ratios (Fig 3B). With NNTCNN, TCN and NTC median ratios close to zero, most of the sequences are not A3-footprinted (Fig 3B, box plots). However, the distribution of the NTC ratios is bimodal in human and non-human primate viruses with a subpopulation of sequences strongly depleted for the NTC codon (Fig 3B, arrows and 4A red part of the distribution plot).

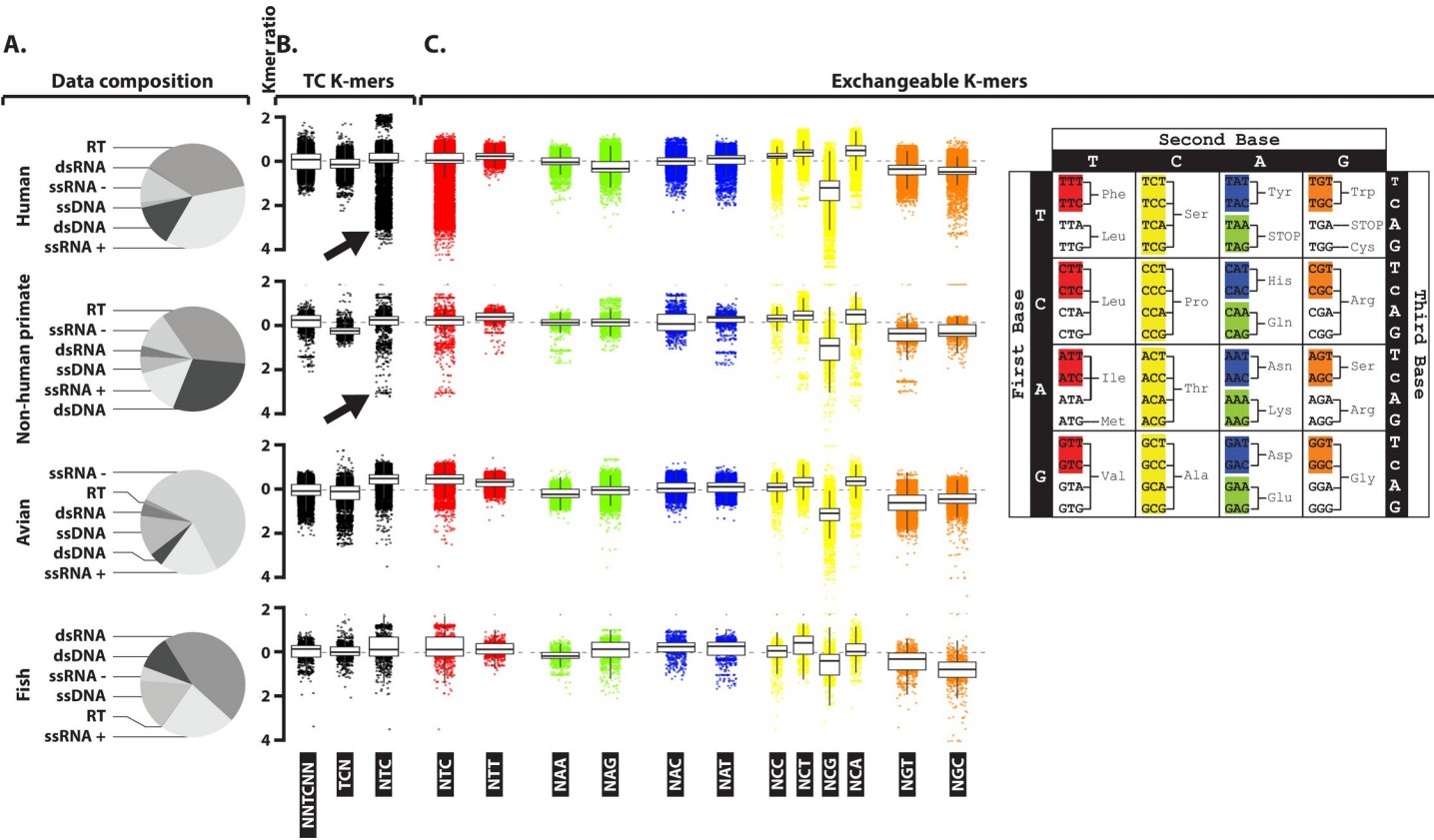

**Fig 3. A sub-population of Human and non-human primate viruses is depleted in NTC codon.** Four datasets including Human viruses (n = 33,400), non-human primate viruses (n = 1,397), avian viruses (n = 9,160) and fish viruses (n = 570) have been analyzed for their observed/expected K-mer ratios. A. The composition of each data set regarding the breakdown into viral groups is illustrated by pie charts. B. The observed/expected ratios of TC dinucleotide at various codon positions for Human, non-human primate, bird and fish viruses are illustrated by dot plots (one point represents one unique viral sequence). C. K-mers are grouped and colored according to their capacity to encode a common amino-acid (in red for NTT/C, in yellow for NCC/G/T/A, in orange for NGT/C, in blue for NAC/T and in green for NAA/G).

Crucially, such footprinted subpopulation is not detected in avian or fish viruses (Fig 3B). Hence, the absence of a detectable A3 footprint in avian and fish viral sequences is consistent with the restriction of the A3 genes family to the mammalian class [37]. It is worth mentioning that each Baltimore's group is represented in the human, non-human primate, avian and fish viral sequence data sets, albeit in different proportions (Fig 3A).

Due to the redundancy of the genetic code, different codons can encode for the same amino acid. The third position of the codons is highly reiterated (redundant) and allows synonymous substitutions. This is notably the case for the NTC and NTT codons where the C or T at the third position are perfectly interchangeable (Fig 3C, colored in red). While we observe a subpopulation of sequences depleted for NTC, such depletion is not observed for NTT. Hence, the NTC depletion cannot simply be explained by the under-representation of an amino acid. The pair NTC/NTT is not the only interchangeable pair. The NAA/NAG, NAC/NAT, NGT/NGC duos and the NCC/NCT/NCG/NCA quartet are also interchangeable. The distribution of NTC ratios remains the sole being bimodal with a subpopulation of strongly depleted sequences. The general NCG depletion (monomodal distribution with a median significantly less than zero) is the result of the well characterized CG dinucleotide under-representation in viral genomes [38]. This depletion is shared in all viral datasets while the NTC depletion is specific to a subpopulation of human and non-human primate viral sequences.

By breaking down the human viruses into their respective Baltimore's group (S3 Fig), we observed that NTC depletion is not present in reverse transcribing nor in negative sense single strand RNA viruses. A mild general depletion is present in double strand RNA viruses. Importantly, a strong general depletion can be observed in double strand DNA viruses. Finally, in single strand DNA and positive sense single strand RNA viruses, certain specific viral sequences appear also significantly depleted. We also observed a mild general NCC depletion in single strand DNA and double strand RNA viruses, justifying further investigation for a possible A3G-induced footprint (S1 Fig). No NCC depletion is observed in double strand DNA, single strand RNA nor in reverse transcribing viruses.

## Screening for human viruses' genomes marked by an A3 footprint

In order to identify A3-footprinted viruses, we detailed the NTC and NNGANN ratios for 870 human viral species (Fig 4A). We observed that the NTC and NNGANN distributions are bimodal with a subpopulation of depleted sequences in each case. We consider a viral species as footprinted by A3s when its NTC or NNGANN median ratio is inferior to the population median by at least two times the standard deviation. Hence, about 17% of the viral species are depleted for NTC (143 species over 870) and about 16% are depleted for NNGANN motifs (136 species over 870). In total, 175 species (22%) present an A3 footprint on either one or both strands. This subgroup is essentially composed by double-stranded DNA viruses with numerous *alpha-*, *beta-* and *gamma- papillomaviridae* (αPV, βPV and γPV) but also *beta-polyomaviridae* (BKPyV, JCPyV, KIPyV, WUPyV and HPyV9) and the delta-polyomavirus MWPyV (Fig 4B). These viruses show a strong depletion for both the NTC and NNGANN motifs by comparison to NNTCNN/TCN and GAN/NGA (Fig 4B). Of note, NTC depletion generally goes with a mild to a significative NTT enrichment (S4 Fig). In the strongly NTC-depleted viruses HPV16, HPV18 and HPV31, a TC depletion is also observed in the non-coding region of the genome regardless of the analyzed motif (S5 Fig). To recapitulate, the A3 footprint on the *papillomaviridae* and *polyomaviridae* is genome-wide and on both strands (Fig 2, Fig 4B and S6 Fig). Importantly, we also identified the single-stranded DNA virus erythroparvovirus B19 and the single-stranded RNA virus HKU1 beta-coronavirus as strongly footprinted by A3s (Fig 4B, highlighted). We will further detail the A3 footprint of these viruses in the following sections.

In order to specifically look for A3G-footprinted viruses, we calculated the NCC and NNGGNN ratios for 870 human viral species (S1B Fig). We observed that the NCC and NNGGNN ratios are mostly narrowly distributed around the zero value. Viruses depleted for NCC are generally single stranded DNA viruses (S1B Fig). However, in many of them we observed a concomitant depletion of the NNCCNN motif casting doubts on the causal link between the observed NCC depletion and A3G editing (S1C Fig).

## B19 erythroparvovirus genome bears a strong A3 footprint

One of the most A3-footprinted virus is the B19 erythroparvovirus. Among the *parvoviridae* family, the TC ratio analysis showed a strong NTC depletion for erythroparvovirus B19 and to a lesser extent for parvovirus 4 and bocavirus 4 (Fig 5A). We also observed a significant depletion of the NNGANN K-mer for each autonomous parvovirus (erythroparvovirus, parvovirus 4 and bocaparvovirus 1, 2, 3 and 4) (Fig 5A). Thus, autonomous parvoviruses appear to be footprinted either on both strands as for the erythroparvovirus, the parvovirus 4 and bocaparvovirus 4 or only on the template strand for the bocaparvovirus 1, 2 and 3. It is interesting to note that the AAV-1 (adeno-associated dependoparvovirus 1) shows a totally different pattern with even a slight enrichment in NTC codons. This dependoparvovirus is not footprinted by the A3s.

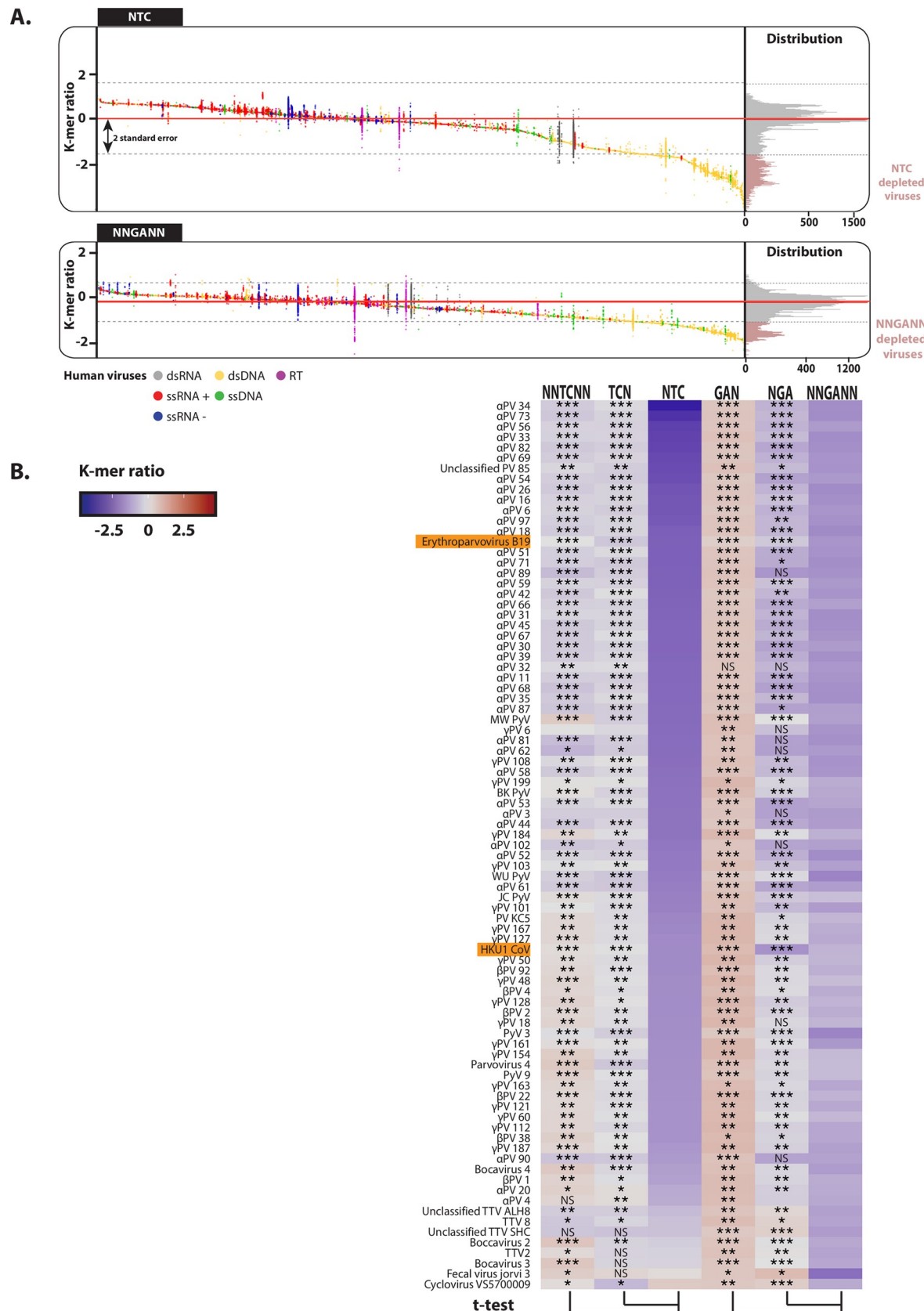

**Fig 4. Search for the A3-footprinted human viruses.** A. The NTC and NNGANN observed/expected ratios for 33,400 human viruses' genomes (from 870 unique species) were calculated, grouped by species and colored according to the Baltimore classification. Each point represents a unique viral genome. Abundance distribution is depicted by a histogram on the right-hand side of the panel. Viral species with an NTC or NNGANN ratio below two times the standard deviation (dotted grey line) from the population median (red line) are the putative A3-footprinted viruses. B. The observed/expected ratios of TC dinucleotide at various codon positions and on both strands (i.e. NNTCNN, TCN, NTC, GAN, NGA and NNGANN) were calculated for the putative A3-footprinted viral species and depicted by a heatmap. A colored scale with increasing shades of blue indicating depletion and increasing shades of red indicating enrichment. P-values were calculated by Student's unpaired, two-tailed t-test (NS for not significant, * p< 0.05, ** p< 0.01, *** p< 0.001). (PV stands for papillomavirus, PyV for polyomavirus).

Detailed analysis of the erythroparvovirus B19 sequences shows a nearly complete NTC cleansing along the whole genome (Fig 5B, red marks). On the contrary, NTT codons are distributed all along (Fig 5B, green marks). Some NTC codons remain present in a short, discrete section of the NS1 gene. This region also encodes for the 7.5k protein in another coding frame. Hence, the remaining TC motifs in the NS1 gene are TCN codon context in the 7.5k protein

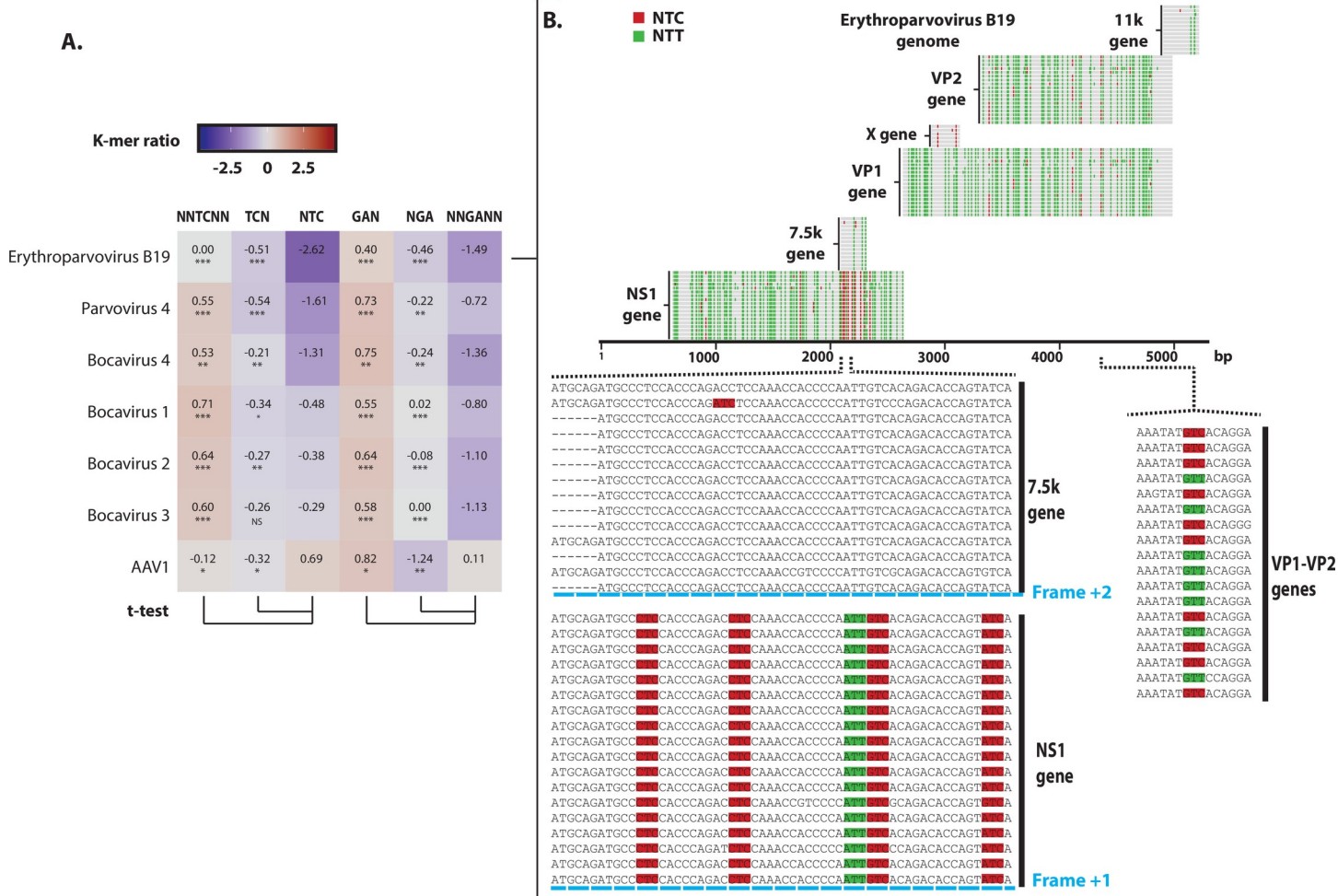

**Fig 5. Intensive A3 footprint on both strands of the B19 Erythroparvovirus genome.** A. The observed/expected ratios of TC dinucleotide at various codon positions for the B19 Erythroparvovirus were compared to those of the other human members of the *parvoviridae* family and depicted by a heatmap. A colored scale with increasing shades of blue indicating depletion and increasing shades of red indicating enrichment. P-values were calculated by Student's unpaired, two-tailed t-test (NS for not significant, * p< 0.05, ** p< 0.01, *** p< 0.001). B. Coding sequences (NS1, 7.5k, VP1, X, VP2 and 11k) from 18 full-length B19 erythroparvirus were depicted by grey lines overlaid by red marks to symbolize NTC and green marks to position NTT codons. Zoom-in detailed a 60 bp-long sequence from the NS1 and 7.5k genes (from nucleotide 1723 to 1783). A second zoom-in detailed a 15 bp-long sequence from the VP1-VP2 genes (from nucleotide 3973 to 3987).

gene. The mutation of those TCs would introduce non-synonymous mutation in the 7.5k protein. This probably explains the conservation of those TC motifs.

Moreover, among the 18 sequences illustrated in Fig 5B, some locations harbor a mix of NTC and NTT codons, suggesting that C to T transition is still an active process (zoom for VP1-VP2 sequence). An A3 footprint can also be observed in the template strand as shown by the NNGANN depletion (Fig 5A and S7 Fig). These observations show that the B19 erythroparvovirus is submitted to an ongoing and strong A3 selection pressure acting on both strands of the virus.

### Human endemic coronaviruses but not zoonotic coronaviruses carry an A3 footprint

By comparing the NTC ratio to the NNTCNN and TCN ratios, we observed a common NTC depletion in the NL63, 229E, HKU1 and OC43 coronaviruses; the HKU1-CoV being the most strongly deleted in NTC codons (Fig 6). In coronaviruses, all viral genes are encoded by the positive strand. In other words, the coding strand of each gene is on the positive strand. Therefore, the depletion of NTC codons is indicative of an A3 activity on the positive strand. These observations corroborate the *in vitro* detection of a soft rate of A3C, A3F and A3H editing on the NL63 genome and the NNU/NNC codon bias previously reported for the HKU1 coronavirus [19,34].

Next, we investigated the presence of an A3 footprint on the template strand (corresponding to the negative strand in coronaviruses) by comparing the 5'GA ratios in different codon contexts. However, we did not observe a progressive depletion of the GA motif (i.e. NNGANN ratio < NGA ratio $\leq$ GAN ratio) which would be expected in the presence of a GA to AA mutational pressure. For that reason, we cannot rule on the presence of an A3 footprint on the negative strand (Fig 6).

Finally, unlike endemic viruses, the zoonotic viruses MERS-CoV, SARS-CoV-1 and SARS-CoV-2 and their animal ancestors camel-MERS, bat-MERS and bat-SARS are not depleted for NTC codons (Fig 6).

### Looking for an A3 footprint at the gene level

Since a non-random distribution for A3 mutations has been reported for some viruses, we then looked for spatially circumscribed A3 footprint; i.e. A3 footprint limited to certain viral genes To limit our screening on genes which are depleted compared to the whole genome, we subtracted to the genic K-mer ratio, the corresponding genomic K-mer ratio to define the differential ratio for NTC and NNGANN K-mers (Fig 7A). In other words, we looked for viruses harboring local A3 footprint amongst an otherwise non-footprinted genome. Thus, differences between genic and genomic K-mer ratios were calculated for 252,766 viral genes. Fig 7B shows the viral genes having an NTC (or NNGANN) differential ratio inferior to the median by at least two times the standard deviation. Thus, we identified many genes being footprinted by A3 among otherwise unaffected viral genomes. Most of these genes belongs to two families of double-stranded DNA viruses; i.e. *herpesviridae* (HHV-1, 2, 3, 4, 5 and 8) and *adenoviridae* (AdV A, B, C, D, E and F). We also observed A3-footprinted genes in the reverse transcribing HBV, HIV-1 and HTLV-1. In order to better shed light on the possible mechanisms responsible for such editing, we position the identified genes along the corresponding viral genomes and detailed these analyses in the following sections.

An equivalent analysis was performed to report a local A3G footprint amongst an otherwise non-footprinted genome. NCC and NNGGNN-depleted genes are listed in S1D Fig. Similarly to what has been observed at the genome level, many of the NCC-depleted genes are also depleted for the NNCCNN motif making difficult to ascribe the observed NCC depletion to the sole A3G editing activity.

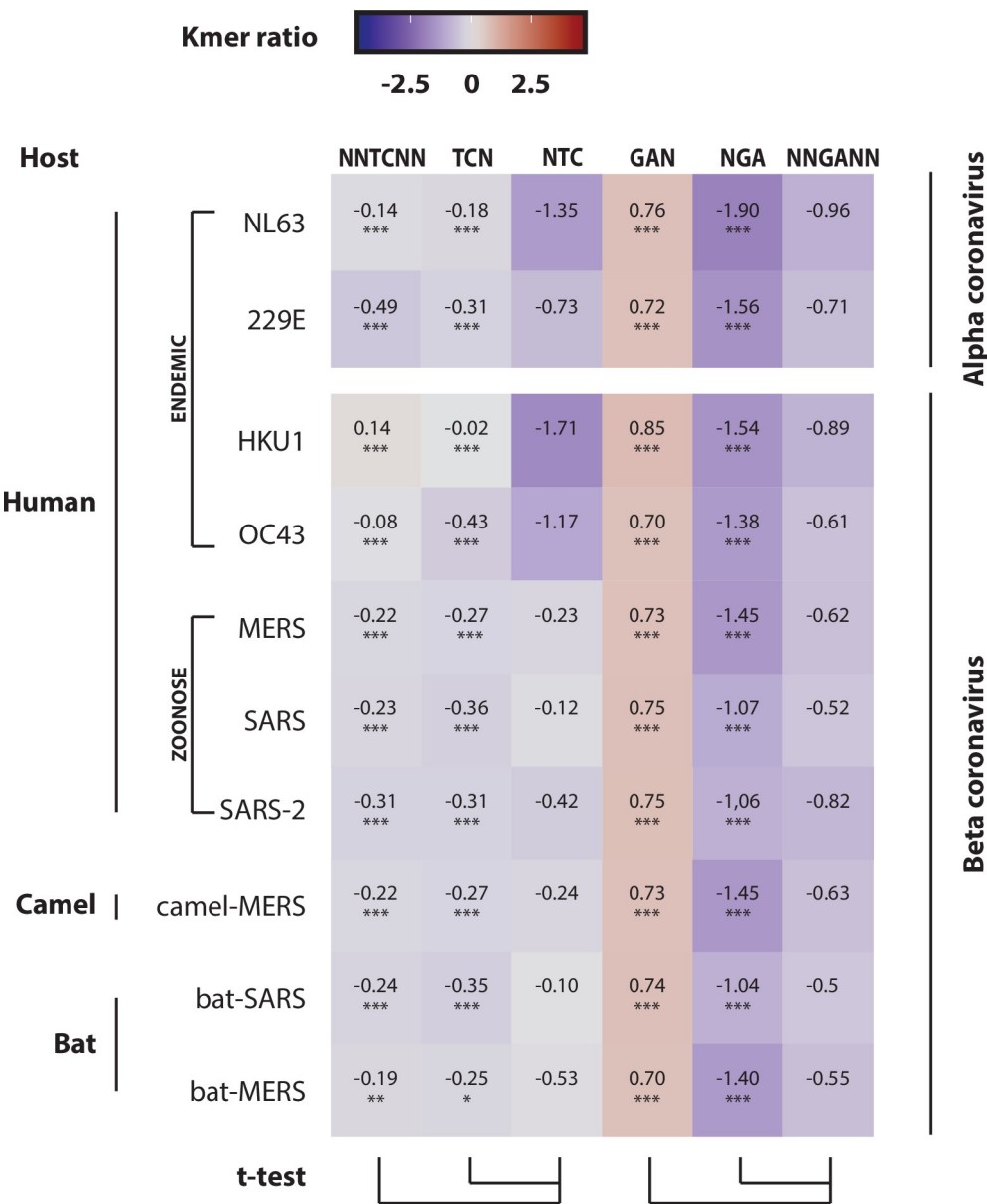

**Fig 6. A3 footprint on endemic but not on zoonotic coronaviruses.** The observed/expected ratios of TC dinucleotide at various codon positions were calculated for endemic human coronaviruses (229E, NL63, OC43 and HKU1) and compared to those of zoonotic coronaviruses (MERS-CoV, SARS-CoV-1 and SARS-CoV-2) and their ancestors (camel-MERS and bat-SARS). A colored scale with increasing shades of blue indicating depletion and increasing shades of red indicating enrichment. P-values were calculated by Student's unpaired, two-tailed t-test (NS for not significant, * p< 0.05, ** p< 0.01, *** p< 0.001).

## Identification of an A3 footprint at the replication origins of adenoviruses and EBV

Adenoviruses A and B present a strong NTC depletion for the E1A and E4 genes (Fig 8A); genes localized at both ends of the linear genome (S8 Fig). The same trend can be observed in Adenovirus C, D and E, although to a lesser extent (S9 Fig). Importantly, these E1A and E4 genes are being strongly depleted for NTC but not for the NNGANN motif (Fig 8B). In other

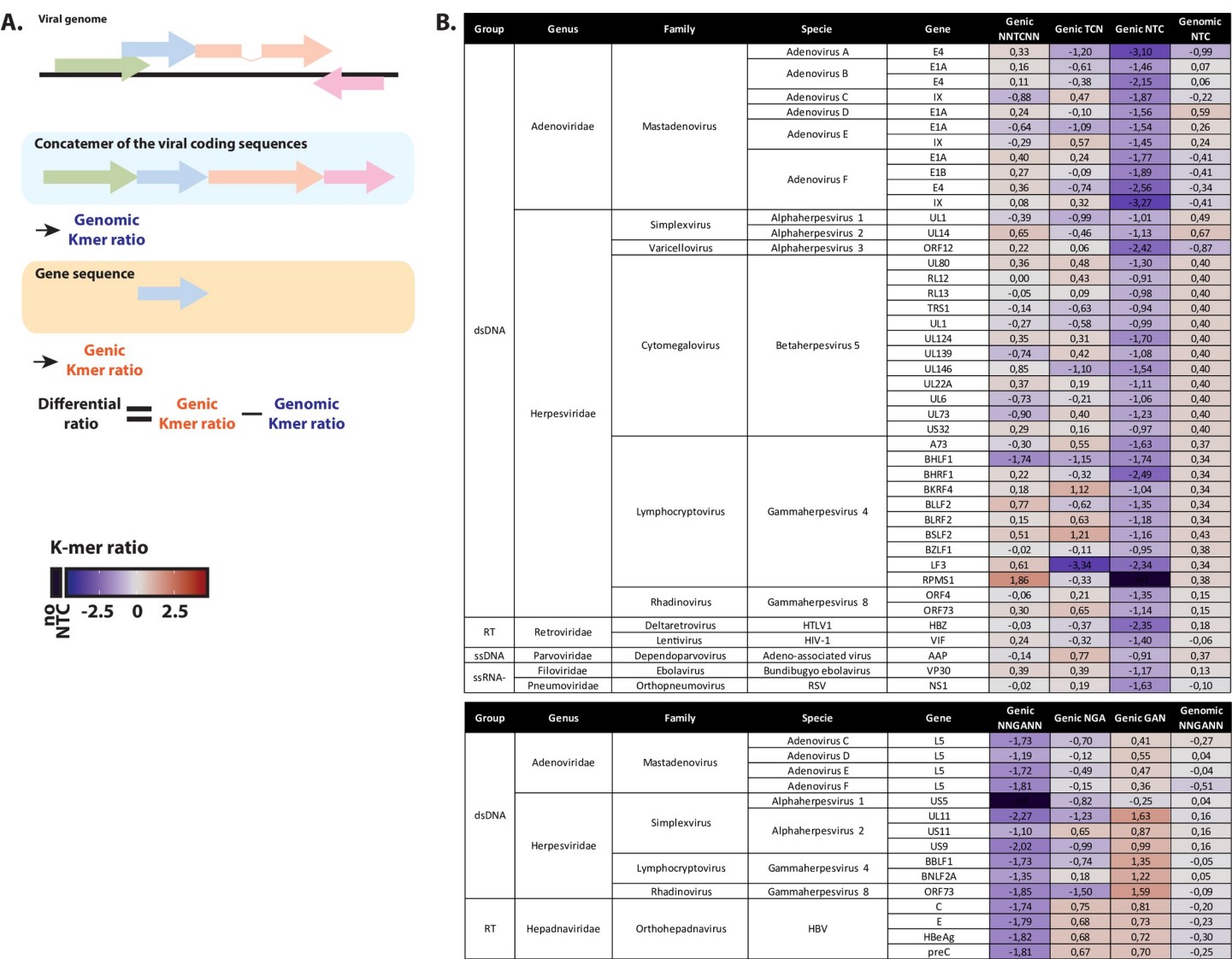

**Fig 7. Search for an A3 footprint at the gene level.** A. Alongside the observed/expected K-mer ratios calculated from the synthetic coding genomes (named genomic K-mer ratios), K-mer ratios were also computed for each viral coding sequence individually (named genic K-mer ratios). Differential ratio is defined as the subtraction of genic K-mer ratio to the corresponding genomic K-mer ratio. B. List of the putative A3-footprinted viral genes and belonging to an otherwise non-depleted viral genome (having at least five reported sequences).

words, these two genes are A3-footprinted on their coding strand only. Due to the relative position of those genes and the strand-displacement strategy used for genome replication, we propose a model where A3 editing would occur specifically during the initiation of genome replication on the displaced strands (Fig 8C). Indeed, at the beginning of DNA replication, the displaced strand corresponds to the coding strand of E1A at one end of the linear genome and to the coding strand of E4 at the other extremity. One might also notice an NNGANN depletion for most of the L5 genes (Fig 8B and S9 Fig). Considering the position and orientation of that gene, such footprint might also reflect an A3 activity on the displaced strand (Fig 8C).

Among the 172 EBV genes, only 12 are significantly depleted for NTC (Fig 7B). Interestingly, the five most depleted are localized around the lytic origins of replication. BHLF1 and BHRF1 are localized on both sides of the first lytic origin of replication and LF3, RPMS1 and

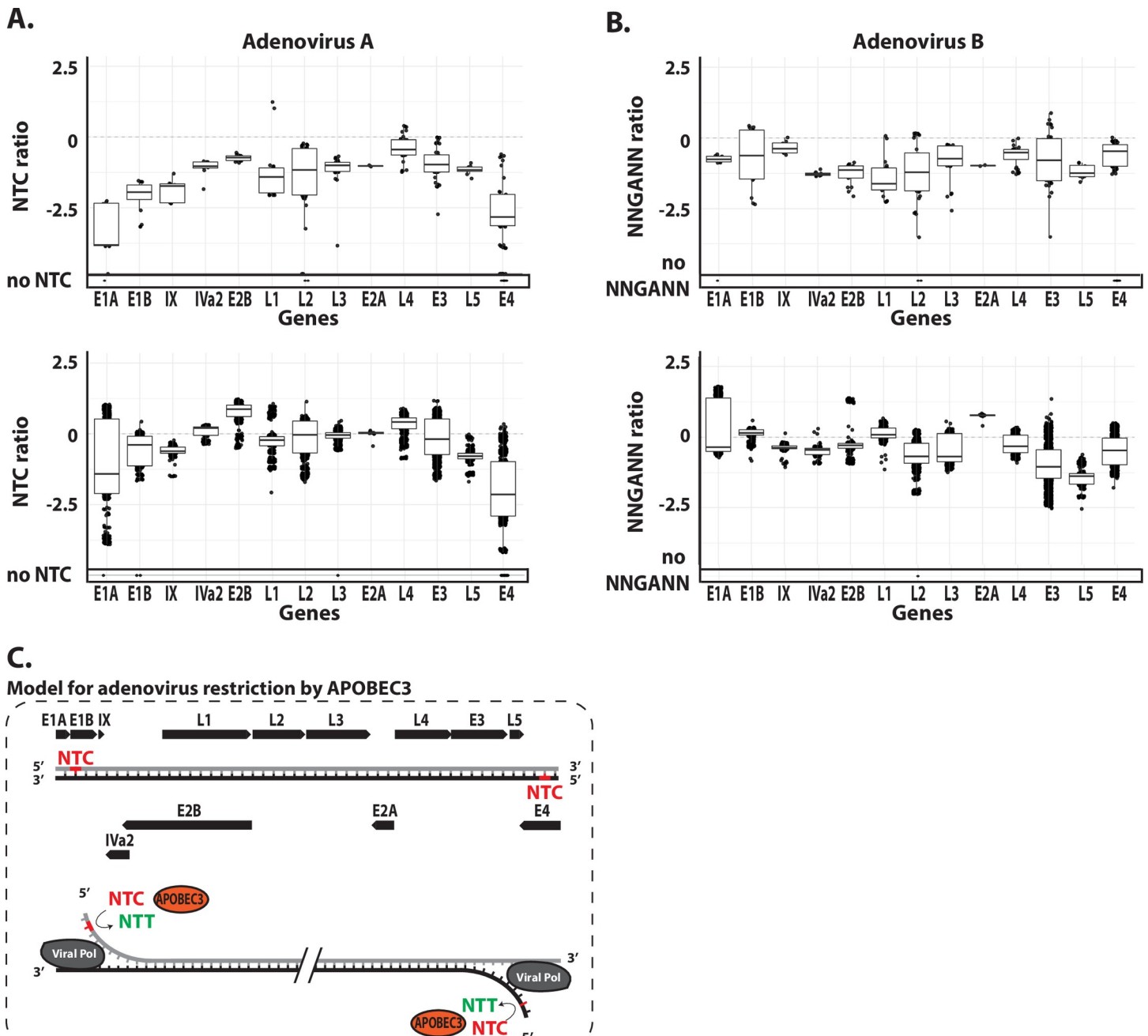

**Fig 8. A3 footprint at the genomic ends of adenoviruses.** NTC observed/expected ratios (panel A) and NNGANN observed/expected ratios (panel B) were calculated for the different genes of the Adenovirus A and B (each point represents a unique coding sequence). C. Proposed model for A3-editing activity on the adenovirus genome. Genes are represented by black arrows. A3-favored NTC sequence is represented in red and the NTT edited product in green.

A73 are on both sides of the second lytic origin of replication (Fig 9B). Similar to the adenoviruses, this local A3 footprint is very much strand-specific and present on the lagging strands of the replication forks surrounding the lytic origins (Fig 9B). Thus, the EBV specific footprint is pointing toward A3 editing during the beginning of replication at the lytic origins. We summarize these observations by a scheme in Fig 9C.

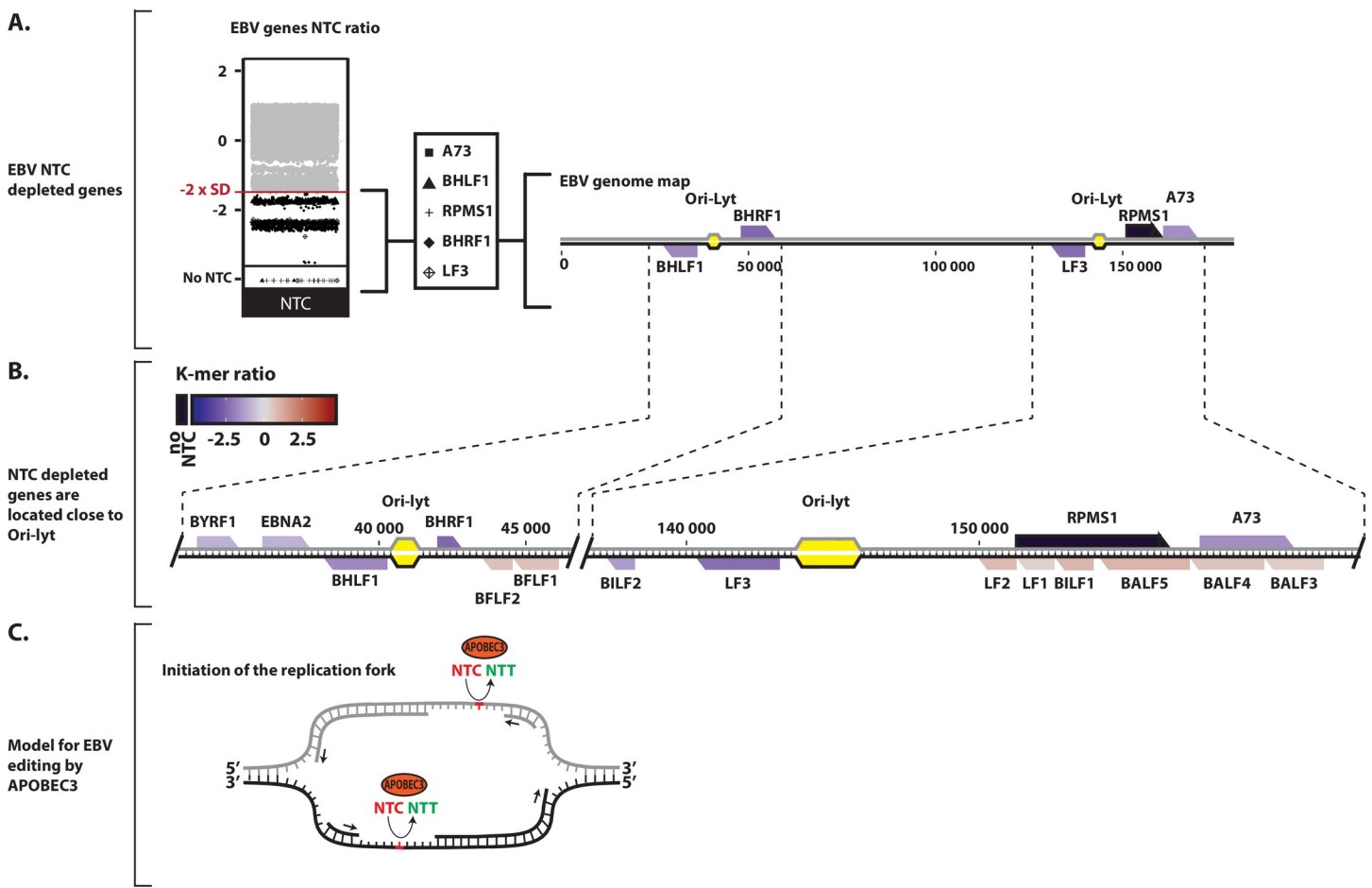

**Fig 9. A3 footprint at the lytic replication origins of EBV.** A. NTC observed/expected ratios were calculated for the different genes of EBV (each point represents a unique coding sequence) and the five most A3-footprinted genes were highlighted and positioned on the EBV genome map. B. Zoom-in detailing the NTC ratios of the genes surrounding the Ori-Lyt (lytic origin of replication) of EBV. A colored scale with increasing shades of blue indicating NTC depletion and increasing shades of red indicating NTC enrichment. C. Proposed model for A3-editing activity favoring the lagging strand at the EBV lytic origin of replication. A3-favored NTC sequence is represented in red and the NTT edited product in green.

## The footprint on the HTLV-1, HBV, HIV-1 and HIV-2 genomes fits with editing during reverse transcription

We observed that the HBZ gene of the HTLV-1 virus is depleted for NTC codons (Fig 7B). Because HBZ is an antisense transcript, its coding strand corresponds to the genomic negative strand. Hence, the NTC depletion of the HBZ gene is indicative of an A3 editing activity on the negative strand (Fig 10A and 10B). We then wondered whether such A3 footprint was restricted to the HBZ coding region or rather extend further. We observed that the coding sequences of the sense transcripts Gag, Pro, Pol and Tax are depleted for the NNGANN motif (Fig 10A and 10B). These observations suggest that A3s left an evolutionary footprint on the HTLV-1 virus through editing during reverse transcription.

Depletion for the NNGANN motif has been observed for the C, preC/HBeAg coding sequences of HBV (Fig 7B). These observations support the involvement of an A3 editing activity on the DNA negative strand during the reverse transcription process. However, it is interesting to report that nor the Pol neither the S and X coding sequences are being foot-printed (Fig 10C and 10D)

Conflicting data were reported concerning the presence of an A3 evolutionary footprint on the HIV-1 genome [31,32]. Fig 11 reports ratios of the TC and GA-containing K-mers for the HIV-1, HIV-2 and SIV genomes. The HIV-1 genomes were spread out into their respecting groups (M, N, O) and subtypes (group M subtypes A, B, C, D and E). No NTC depletion was observed on the HIV-1, HIV-2 and SIV genomes (Fig 11). We concluded that A3s did not leave a footprint on the plus strand. Importantly, a mild but consistent NNGANN depletion was observed in HIV-1, HIV-2 and SIV genomes, depletion compatible with A3-editing during reverse transcription.

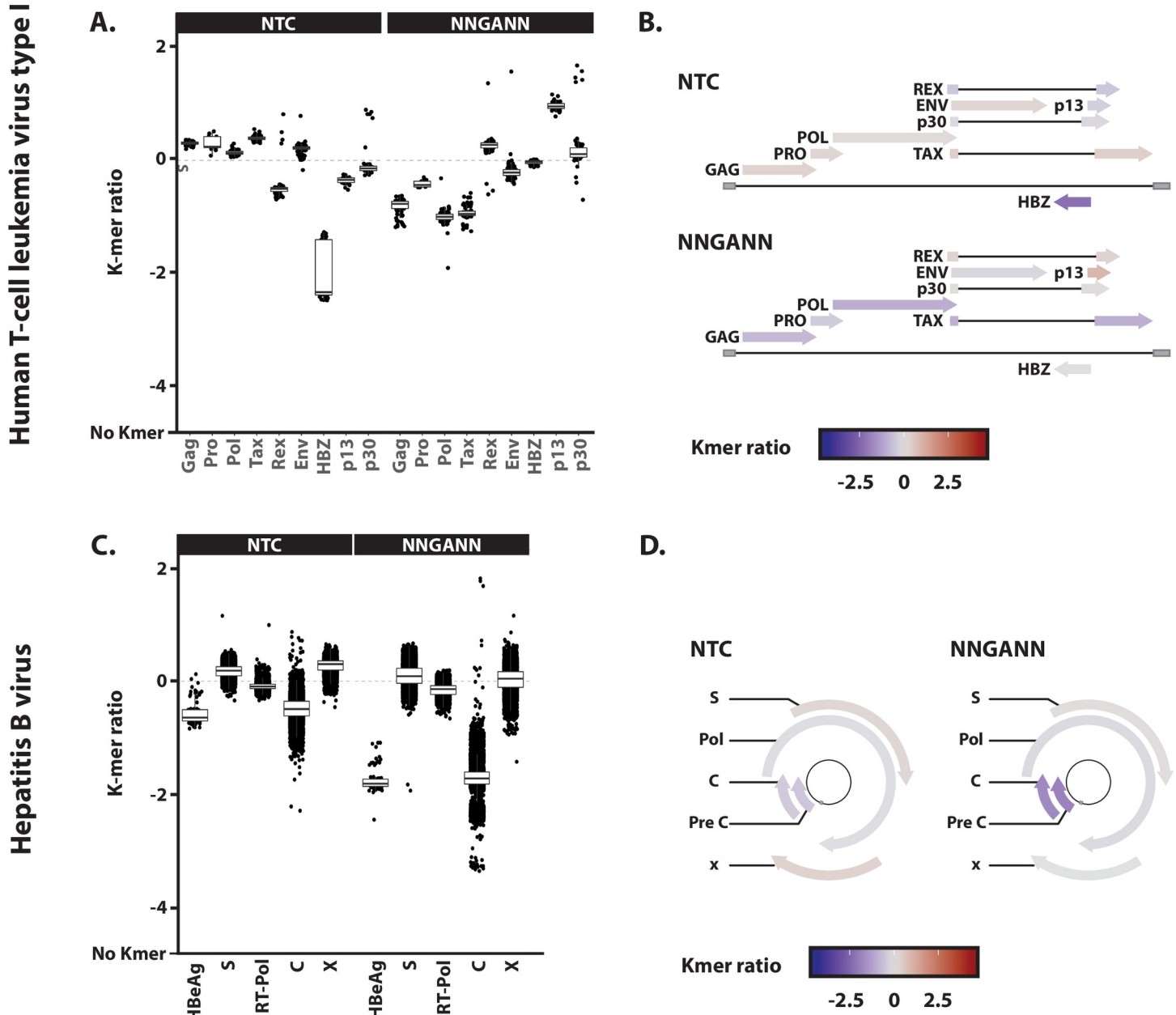

**Fig 10. A3 footprint on the negative strand of HTLV-1 and HBV.** A. NTC and NNGANN observed/expected ratios were calculated for the different genes of HTLV-1. B. Each gene specific NTC and NNGANN ratio median values were reported on HTLV-1 genome map by a colored scale. C. NTC and NNGANN observed/expected ratios were calculated for the different genes of HBV. D. Each gene specific NTC and NNGANN ratio median values were reported on HBV genome map by a colored scale.

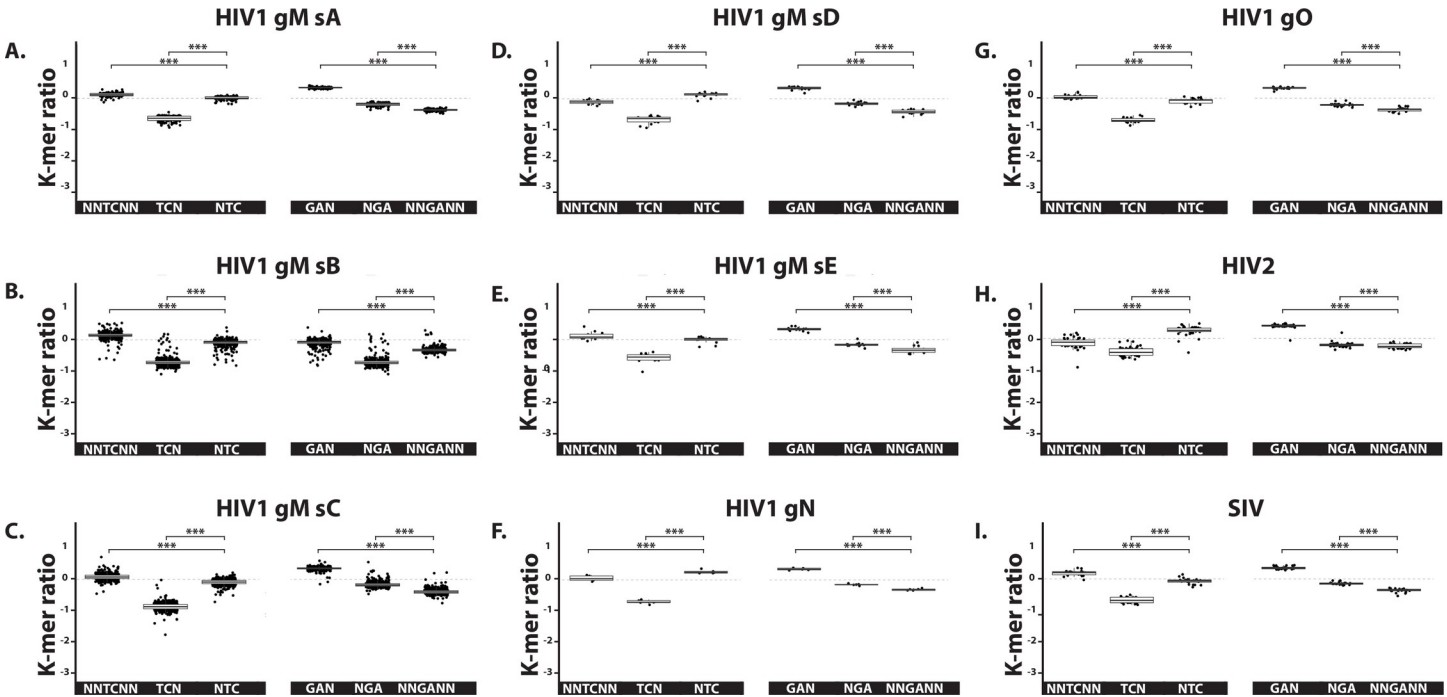

**Fig 11. A3 footprint on the negative strand of HIV-1, HIV-2 and SIV.** The observed/expected ratios of TC dinucleotide at various codon positions and on both strand (i.e. NNTCNN, TCN, NTC, GAN, NGA and NNGANN) were calculated for the genomes of HIV-1 (distributed into their respective groups and subtypes, panels A to G), HIV-2 (panel H) and SIV (panel I). Each point stands for a unique full-length viral genome. Median and quartile are depicted by a boxplot. P-values were calculated by Student's unpaired, two-tailed t-test (NS for not significant, * $p < 0.05$, ** $p < 0.01$, *** $p < 0.001$).

### Search for AID or APOBEC1-footprinted viruses

The APOBEC family of genes also counts the AID (or AICDA), APOBEC1, APOBEC2 and APOBEC4 genes. AID is critical for somatic hypermutation and class switch recombination by editing the immunoglobulin loci in B cells [39]. APOBEC1 plays an important role in lipid metabolism by editing the apolipoprotein B pre-mRNA [40,41]. Importantly, APOBEC1 and AID appear also to participate to the restriction of viruses and retroelements [1,42]. Evidence for AID and APOBEC1 evolutionary footprints were investigated by looking for the depletion of their favored motifs, respectively WRC for AID [43] and WCW for APOBEC1 [44].

The distributions of the WRC and NNGYWN ratios do not point towards viruses significantly footprinted by AID at the whole genome level (S10B Fig). Nevertheless, putatively AID-footprinted genes were identified in several double strand DNA viruses, notably the B-cell-infecting virus EBV (S10C Fig).

The distributions of the NWCWNN and NWGWNN ratios show evidence of genome-wide footprinted viruses (S11B Fig). However, it is not possible to disentangle the APOBEC1 footprint from the APOBEC3 footprint as the 6-mers NWCWNN contains the 3-mers NTC. The putatively APOBEC1-footprinted viruses are those that also bear the putative APOBEC3 footprint (Fig 4B and S11C Fig).

Of note, APOBEC2 and APOBEC4-favored motifs have not been described so far.

### Discussion

In this study, we investigated the distribution of the A3 footprint along a large set of 33,400 human virus complete genomes. We first observed that no less than 22% of all referenced

human viral species have a genome-wide A3 footprint. Among these, we mainly identified viruses from the *papillomaviridae*, *polyomaviridae*, *coronaviridae* and autonomous *parvoviridae* families. In addition to this category of viruses targeted over their entire sequence, we have identified viruses which have an A3 footprint spatially limited to a short section of their genome. This is notably the case for certain *herpesviridae* and *adenoviridae* where the A3 footprint is localized on genomic sequences used to initiate replication of viral DNA.

Our study is in line with previous publications reporting the presence of an A3 footprint on *papillomaviridae* [29] and on the BK polyomavirus [17]. Above all and because we analyzed all currently annotated human viruses with the same approach, we can compare the magnitude of the A3 selection pressure between different viral families. Thus, we show that the *papillomaviridae* and the *polyomaviridae* families are those whose footprint is most intensive (Fig 4A). Those viruses have evolved to thrive under an ongoing and strong A3 selection pressure. The strong NTC depletion reduces exposure of the viral genome to the introduction of uracil and consequently to the base excision repair-mediated DNA degradation. Importantly, not only do these viruses tolerate such pressure, but they even actively promote the expression of certain A3 proteins. Indeed, high risk α-HPVs have been shown to trigger and stabilize A3A and A3B via their oncoproteins E6 and E7 [16,45–47]. Likewise, it has recently been shown that BK and JC β-PyV upregulate A3B through their large T antigen [17,48]. In both the α-HPVs and β-PyVs, the induced A3 proteins are enzymatically active and therefore capable of deamination [17,49]. The selective advantage which would provide a sustained expression of A3A and/or A3B proteins is still debated. On the one hand, A3A has been shown to restrict HPV *in vitro* [45]. Those viruses are still susceptible to A3 restriction. Indeed, deamination of the remaining TC motifs are most of the time non-synonymous. On the other hand, the deaminase activity could positively impact viral fitness by participating to the genetic diversification of the virus or even by protecting the host cell against the reactivation of retroelements [50,51]. We speculate that the error rate of the host DNA polymerase could be too low for viruses with such small DNA genome, hence requiring the A3 editing activity to drive their evolution. Within the *polyomaviridae* family, the magnitude of the A3 footprint differs significantly between species; species of the *betapolyomavirus* genus appearing to be the most strongly footprinted (S2 Fig). Such differences could find their origin in the capacity of the large T Ag at inducing the A3 proteins. To draw a parallel with the *alpha-papillomaviridae*, E6 from high-risk α-HPVs were found to be more potent at inducing A3B than those of low-risk strains [49]. Besides, the cell type hosting the virus can also influence the level of A3 expression. The difference in tissue tropism between the *alpha-* and *beta-papillomaviridae* has been proposed to explain the stronger footprint on the former [29]. The full spectrum of tissue and cell tropism has not been clearly established for the *polyomaviridae*, making this type of correlative analysis tricky. Our analysis also shows that the A3 footprint is present on both strands of the *papillomaviridae* and *polyomaviriridae* genomes. This is compatible with an editing activity during viral DNA replication.

Among the *parvoviridae* family, the erythroparvovirus B19 exhibits an intensive footprint on both strands of its genome, the bocaparvoviruses being mainly footprinted on the negative strand and the dependoparvovirus adeno-associated virus-1 showing no evidence of A3 selection pressure. These dissimilarities might be explained by differences at the replicative and packaging levels. Thus, the *parvoviridae* family consists of viruses that package a single copy of their short linear single-stranded DNA genome into preformed capsids. The packaging takes place in the nucleus of the infected cell. While most can encapsidate DNA strands of either polarity with equal efficiency, some family members, predominantly package negative strand genome. In the case of the erythroparvovirus, there is an equivalent amount of positive and negative genome that is produced during replication and subsequently encapsidated (reviewed

in [52]). For *bocaparvoviridae*, the replication produces 90% of negative ssDNA [53]. Such difference could explain the location of the A3 footprint in the negative strand of the *bocaparvoviridae*. Thus, we propose that A3 editing activity takes place inside the nascent virions of the autonomous *parvoviridae*.

Our screening also reported the *coronaviridae* as A3-footprinted. The canonical substrate for the A3 proteins is single stranded DNA and until recently, the viruses identified as being restricted by the A3 deaminase activity were either DNA viruses or viruses having a DNA intermediate (i.e. reverse transcribing viruses). However, recent reports demonstrated that A3A and A3G can deaminate ribocytidine within a single stranded RNA molecule [54–56]. *In vivo*, A3 mutational signature has been reported in the positive single strand RNA Rubella virus [57]. Importantly, Milewska and colleagues demonstrated that cytoplasmic A3s can restrict the NL63 coronavirus *in vitro* [19]. The A3-mediated restriction of the HCoV-NL63 appears to be both deaminase-dependent and independent. A3 restriction did not cause hypermutation on the viral genome, but C to T and G to A point mutations were observed in HCoV-NL63 viruses passaged in A3-expressing cells but not in wild-type cells. It is a matter of debate whether the hypermutated genomes could not be retrieved because of the high fitness cost of such mutations or because the A3 are less processive on coronaviral RNA. Additionally, A3 proteins have been shown to interact with the nucleoproteins of the HCoV-229E, HCoV-NL63 and SARS-CoV-1 viruses [19,58]. Finally, a recent report demonstrates the presence of APOBEC and ADAR editing on the SARS-CoV-2 transcriptome [59]. Thus, knowing that A3s can bind the nucleoprotein and that A3 footprint is present on the positive strand of the viral genome, we suggest that A3 editing occurs on the packaged genome. Two *beta-coronaviridae* are endemic to humans (HCoV-OC43 and HCoV-HKU1), they are widespread, have been circulating in human for at least several decades and may cause 10 to 15% of common colds (review in [60]). Both have an A3 footprint on the positive strand. In comparison, no evidence of footprint was observed on the zoonotic *beta-coronaviridae* SARS-CoV-1, MERS-CoV or SARS-CoV-2. The absence of an evolutionary footprint on SARS-CoV-1 and MERS-CoV could find its explanation in the relative low number of infected individuals and the short duration of viral circulation. According to the World Health Organization, SARS-CoV-1 infected about 8.000 people over a period of few months and have been declared eradicated in May 2004. The MERS-CoV infected so far less than 3.000 people by causing episodic outbreaks in the Middle East. The figures for the SARS-CoV-2 are radically different with more than 5 million confirmed cases as of May 2020. In that respect, it will be interesting to track the evolution of the pandemic SARS-CoV-2 regarding a possible introduction of an A3 footprint through its interhuman transmissions. It is worth reiterating that no footprint was detected on the SARS bat isolates, although the bat A3 locus is the largest and most diverse known repertoire of A3 genes in mammals [61]. Perhaps SARS-like viruses possess a yet unknown and unique A3 inhibiting mechanism. Interestingly, the SARS-CoV-2 genome contains a novel ORF, called ORF10. ORF10 encodes a protein that has been demonstrated to interact with the CUL2 complex [62]. This interaction is reminiscent of the interaction between the Vif of BIV (bovine immunodeficiency virus) and CUL2 [63]. One could speculate that ORF10 could play a Vif-like role in bat and also in Human. Also, a shorter version of SARS-CoV-2 ORF10 is present in all SARS-like viruses [64]. This could explain why the SARS-like viruses are not A3-footprinted.

In addition to this category of viruses that are footprinted on their entire sequence, we identified viruses that show an A3 footprint only on a very limited section of their genome. This is notably the case for the *gamma-herpesviridae* EBV and *adenoviridae*. The A3 footprint on EBV is spatially limited to the lagging strands around the lytic replication origins. Interestingly, both EBV and KSHV were recently demonstrated to encode viral proteins capable of

inhibiting A3B activity [28,65]. Those viral proteins (EBV BORF2 and KSHV ORF61) are both the large subunit of the ribonucleotide reductase, expressed during lytic replication and providing the precursors necessary for viral DNA synthesis. We speculate that their expression could avoid the extension of the A3-editing further along the viral genome. Of course, these viral proteins inhibiting A3B may not be the sole actors protecting the viral genome. The coating of the viral DNA by the major DNA binding protein (DBP), the compartmentalization of the viral DNA replication [66] and the switch from theta to the rolling circle replication might also limit access to the viral single-stranded DNA. The fact that no footprint was detected around the latency origin of replication, ori-P, points toward an A3-editing acting during lytic replication. The strategy deployed by EBV and KSHV to cope with A3 restriction is somehow opposite to the one used by the papilloma and polyomaviruses. Indeed, EBV and KSHV actively protect their genome from the A3s as opposed to the papilloma and polyomaviruses which simply cope with a high mutational rate. We speculate here that the large size of the herpesvirus genome would not tolerate an unrestrained A3 activity. Finally, we identified several AID-footprinted genes in the EBV genome which supports recent observations made by Martinez et al. [33]. Those genes are not spatially clustered and further investigation would be necessary to link the detected footprint to AID activity.

Similarly, we identified an A3 footprint on *adenoviridae* localized at the ends of their linear genome where the origins of replication are located. The presence of an A3 footprint on the lagging strand of the origins of replication and its absence on the leading strand is striking. It parallels the footprint on the lytic origins of replication in EBV and it is also reminiscent of the A3 activity in cancer genomes. Thus, A3-related mutations in cancer genomes are strongly enriched on the lagging strand and early-replicating euchromatic regions [67–70]. The leading strand, being protected by the nascent complementary DNA, is less or even not accessible for deamination. Another circumstance where the viral DNA is transiently single-stranded is during transcription. Indeed, the coding strand within the transcriptional bubble is temporally single stranded. In cancer genomes, the observed distribution of APOBEC3-signature mutations is transcription independent [70]. Again, it is very similar to our observations on viral genomes where no evidence points toward A3-editing during viral transcription. Overall, the similarities between *adenoviridae* and *herpesviridae* regarding their relationship to A3s appear substantial. Both are large double-stranded DNA viruses replicating their genome in the nucleus and capable of lytic and latent/persistent phases. It would not be surprising to find that *adenoviridae* are also able to inhibit A3 activity. Along with these speculations, we found important to underline that the dependoparvovirus AAV-1 is the only member of the *parvoviridae* family which does not have an A3 footprint. It would be interesting to test whether it is an intrinsic characteristic of the satellite virus or whether it is mediated by the helper virus (usually an adenovirus or a herpes virus).

The detection of an A3 footprint on the negative strand of the C (Core) and preC/HBeAg coding sequences of HBV is compatible with A3-editing on the DNA negative strand during reverse transcription. A3-related mutations have been detected on the negative strand of the C and preC region and it has been proposed that these mutations could be beneficial for the virus [71]. Indeed, during the natural course of HBV infection, the HBeAg expression is being lost after production of antibodies against it. HBeAg is an accessory non-particulate protein encoded by the preC mRNA and displaying immunomodulatory properties. HBeAg is described as a tolerogen that allows the virus to establish infection. Seroconversion against the HBeAg leads to the selection of HBeAg-negative mutants. The ability to develop mutations, altering HBeAg expression, can influence the length of the HBeAg-positive phase, which is important for determining the clinical course (reviewed in [72]). Our observation of an A3 footprint in the preC and C region further supports the idea that A3 can positively participate

to the immune escape. That would not be the first example of the hijacking an antiviral weapon for the benefit to the pathogen. Hepatitis D virus is a circular complementary single-stranded RNA virus that requires editing of its genome by a cellular adenosine deaminase (ADAR-1) to complete its life cycle (reviewed in [73]). Likewise, it is striking to note that only the C and preC/HBeAg coding regions are being footprinted and not the others coding sequences downstream. In fact, the mutational load introduced by the A3s is much stronger on the 5' end of the negative strand (corresponding to the Pol, S and X regions) because the newly synthesized double strand DNA eventually displaces the single strand DNA from the capsid walls, making it accessible to deaminase activity [74]. Thus, the 5' end of the negative strand is found to be frequently hypermutated; the term hypermutation referring to as mutations clustered on a short sequence. It is essential to underline that our observations reflect the A3-induced mutations which were conserved and not those which put an end to the viral cycle. Consequently, the phenomenon of hypermutation will not leave an evolutionary footprint. In this respect, the HBV mutation spectrum from *in vivo* cirrhotic samples shows that even though the majority of HBV genomes are strongly mutated by A3s and these virions are probably defective, a small fraction is slightly modified and may therefore still be infectious [71]. Finally, the Pol, S and X genes overlap on different reading frames, which implies that these coding regions are less permissive to mutations (a silent mutation in one frame may not be in the other frame).

We paid particular attention to HIV in our analysis as the APOBEC3 antiviral activity has been historically discovered in that field of research [6]. We observed a weak A3 evolutionary footprint on the minus strand of the HIV genomes in support of the observations made by Jern et al. [31]. The weakness of the footprint can be explained at least in part by the efficiency of the A3-inhibiting protein Vif. Also, the error-prone RT could be responsible for the reversion of some A3-induced mutations providing that the virus can still complete its life cycle. Finally, A3s are well known to also restrict HIV through a deaminase independent activity, therefore without leaving any footprint.

The intensity of the A3 footprint is very strong in many *papillomaviridae* and *polyomaviridae*. Several viruses of these families are well-known tumor viruses (HPV-16, HPV-18, Merkel cell polyomavirus, etc.) and a mechanistic link between A3 expression and the development of cancer has been established in HPV positive cervical and oesopharyngeal cancers [75,76]. S12 Fig shows oncogenic viruses (confirmed or suspected) and their respective A3 footprint. It illustrates that an A3 footprint is not present in every tumor virus. HCV shows no A3-footprint, is still definitely a tumor virus. Nevertheless, we wonder whether the presence of a strong footprint may suggest involvement in cancer. The BK and JC polyomaviruses have a footprint as intensive as HR-HPVs (S12 Fig). BK PyV infects the kidneys and the urinary tract and is suspected of playing a role in certain bladder cancer where viral expression and integration have been reported [77,78]. BK PyV triggers A3 expression *in vitro* and the A3-related mutations found in bladder tumors account for two-thirds of the total mutational load [17,79]. Similarly, along with the *alpha*, the *beta-papillomaviridae* show a similar A3 footprint (Fig 4B). Some members of the *beta-papillomavirus* genus are suspected to play a role in non-melanoma skin cancer (cutaneous squamous-cell carcinoma) [80,81]. Even so they do not seem to insert into the cell genome, they might promote carcinogenesis initiation. Some β-HPVs were demonstrated to potentiate the deleterious effect of UV radiations and to drive skin carcinogenesis in mice with a hit-and-run mechanism [82]. Finally, the erythroparvovirus B19 is one of the most footprinted virus. While its replication occurs primarily in erythroid tissues, the erythroparvovirus B19 commonly persists in a wide range of tissues [83]. A link between the erythroparvovirus B19 and thyroid cancer has been proposed but evidence is scarce to date (72–74). Of note, the A3-related mutations in thyroid tumors make about 40% of the total mutational

load [79]. We think that further research should be carried out to rule in or out the involvement of these later viruses in cancer.

In conclusion, the present study represents the first global screening for the A3 selection pressure on all currently annotated human viruses. We demonstrate that many *papillomaviridae*, *polyomaviridae*, autonomous *parvoviridae* and *coronaviridae* can thrive despite being under the selective pressure of the A3 proteins. Those viruses cope with A3 editing activity thanks to a deep cleansing of A3-favored motifs in their genome. *Herpesviridae* and *adenoviridae* display a subtler A3 footprint limited to the lytic origins of replication, probably thanks to active mechanisms of A3 inhibition. A3 deamination appears to occur during replication of viral DNA (sometimes limited to the lagging strand) for the double-stranded DNA viruses and/or inside the capsid for the single-stranded DNA and RNA viruses. The causal link established between HPV infection and the A3 mutational signature in human cancer also lead us to propose to consider the *beta-papillomaviridae* and the erythroparvovirus B19 as potentially promoting A3 expression and therefore exposing the cell genome to a mutagenic activity.

## Material and methods

### Fasta sequences

We downloaded complete viral genomes from the "NCBI Virus" database (https://www.ncbi.nlm.nih.gov/labs/virus/vssi/#/) as released in April 2020. We retrieved only full-length genomes by selecting "Complete" for the criterion "Nucleotide Completeness". We retrieved Human viruses by selecting "Humans" for the criterion "Host". We retrieved non-human primate viruses by selecting "Primate" for the criterion "Host" and by deducting the human viruses from this data set. We retrieved avian viruses by selecting "Aves (birds)" for the criterion "Host". We retrieved fish viruses by selecting "Actinopterygii (ray-finned fish)" for the criterion "Host". We also retrieved Camel MERS viruses by selecting "Camelus dromedaries (Arabian Camel)" for the criterion "Host" and "MERS-CoV" for criterion "Virus". We retrieved Bat-MERS viruses by selecting "Chiroptera (bats)" for the criterion "Host" and "MERS-CoV" for the criterion "Virus". We retrieved Bat-SARS viruses by selecting "Chiroptera (bats)" for the criterion "Host" and "Severe acute respiratory syndrome-related coronavirus" for criterion "Virus". The dataset of Human viruses was supplemented by manually curated human virus complete genome sequences from the "NCBI nucleotide" database. Using these criteria, 33,400 Human, 1,397 non-human primate, 9,160 avian, 570 fish, 259 Camel MERS, 5 Bat MERS and 33 Bat SARS full-length viral genomes were collected. GenBank accession ID's are treated as unique and listed in the S1, S2 and S3 Tables.

### Calculation of the K-mer representation ratio

A K-mer encompasses a collection of sequences with a common motif. For instance, the NTC K-mer includes the ATC, CTC, GTC and TTC sequences. In addition, as we limit our analysis to coding sequences, we force our K-mers to be in the reading frame and therefore to correspond to codons. For example, the NTC K-mer actually includes the ATC, CTC, GTC and TTC codons. Following the same logic, the NNTCNN K-mer comprises the 256 pair of codons having a T at the end of the first codon and a C to start the second codon. We calculated the observed vs. expected K-mer representation ratio as described by Warren *et al.* [29]. Briefly, each coding sequence has been randomly shuffled a thousand times, retaining only the nucleotide composition. The expected count of a given K-mer is calculated as the average of the occurrences of this K-mer over the thousand iterations. The K-mer ratio is given as the log2 ratio of the observed occurrence of this K-mer to the expected occurrence. To calculate the ratio of a given K-mer for an entire viral genome, a "synthetic coding genome" was generated

by concatenating the different coding sequences ([Fig 1B]). The synthetic coding sequence is then randomly shuffled a thousand times and K-mer ratio calculated as above. A K-mer ratio $<< 0$ indicates K-mer under representation and a K-mer ratio equal to zero means that no representation bias is observed.

## Statistical analysis

Unpaired Student's t test has been used where appropriate. The results were considered statistically significant at a P-value of <0.05. All boxplot, heatmap and map representations have been generated using ggplot R package.

## Supporting information

**S1 Fig. Search for A3G-footprinted human viruses.** A. A3G favors deamination of cytidine when preceded by another cytidine. The 5'CC dinucleotide motif is depicted in three possible codon contexts on both coding and template strand. Depending on the position of the mutated C, the C to T transition can be synonymous (S) or non-synonymous (NS). Proportion of S and NS mutations is reported when the two types of mutation can be produced. Because synonymous mutations are more likely to be retained, A3G-footprinted viruses should display to a stronger depletion of NCC codons compared to CCN or NNCCNN motifs (and/or a depletion of NNGGNN motifs versus the GNN and NGG motifs). B. The NCC and NNGGNN observed/expected ratios for 33,400 human viruses' genomes (from 870 unique species) were calculated, grouped by species and colored according to the Baltimore classification. Each point represents a unique viral genome. Viral species with an NCC or NNGGNN ratio below two times the standard deviation (dotted grey line) from the population median (red line) are retained for further analysis in panel C. C. The observed/expected ratios of 5'CC dinucleotide at various codon positions and on both strands (i.e. NNCCNN, CCN, NCC, GGN, NGG and NNGGNN) were calculated for the NCC and/or NNGGNN depleted viral species and depicted by a heatmap. A colored scale with increasing shades of blue indicating depletion and increasing shades of red indicating enrichment. P-values were calculated by Student's unpaired, two-tailed t-test (NS for not significant, * p< 0.05, ** p< 0.01, *** p< 0.001). D. List of the viral genes displaying NCC or NNGGNN depletion and belonging to an otherwise non-depleted viral genome.
(PDF)

**S2 Fig. NTC depletion among several *polyomaviridae* family members.** The observed/expected ratios of TC dinucleotide at various codon positions (i.e. NNTCNN, TCN, NTC) were calculated for several polyomaviruses and the corresponding genus (alpha, beta and delta) is reported for each virus.
(PDF)

**S3 Fig. K-mer ratios of the human viruses split according to Baltimore's groups.** Human viruses were broken down into their respective Baltimore's group and analyzed for their observed/expected K-mer ratios.
(PDF)

**S4 Fig. NTT, NTA and NTG K-mers ratios of the A3-footprinted viruses.** The observed/expected ratios of NTC, NTT, NTA and NTG K-mers were calculated for the putative A3-footprinted viral species and depicted by a heatmap. A colored scale with increasing shades of blue indicating depletion and increasing shades of red indicating enrichment. P-values were calculated by Student's unpaired, two-tailed t-test (NS for not significant, * p< 0.05, ** p< 0.01, ***

p< 0.001).
(PDF)

**S5 Fig. TC depletion in HPV non-coding sequences.** The observed/expected ratios of TC dinucleotide at various "codon" positions (i.e. NNTCNN, TCN, and NTC) were calculated for the non-coding sequences of human papillomavirus 16, 18 and 31.
(PDF)

**S6 Fig. A3 footprint on HPV16, HPV18 and HPV31.** NTC and NNGANN observed/ expected ratios were calculated for the different genes of the HPV16, HPV18 and HPV31 and were reported on their genomic maps using a colored scale with increasing shades of blue indicating NTC depletion and increasing shades of red indicating NTC enrichment. Replication origin is illustrated by a black dot and gene transcriptional orientation is symbolized by black arrow.
(PDF)

**S7 Fig. B19 erythroparvovirus genome is depleted for NNGANN K-mer.** Coding sequences (NS1, 7.5k, VP1, X, VP2 and 11k genes) from 18 full-length B19 erythroparvoviruses were depicted by grey lines overlaid by red marks to symbolize NNGANN and green marks to position NNAANN codons.
(PDF)

**S8 Fig. NTC depletion of the E1A and E4 genes of the Adenovirus A and B.** NTC observed/ expected ratios were calculated for the different genes of the Adenovirus A and B and were reported on their genomic maps using a colored scale with increasing shades of blue indicating NTC depletion and increasing shades of red indicating NTC enrichment.
(PDF)

**S9 Fig. A3 footprint on Adenovirus C, D, E, F and G.** NTC and NNGANN observed/ expected ratios were calculated for the different genes of the Adenoviruses C, D, F and G (each point represents a unique coding sequence).
(PDF)

**S10 Fig. Search for AID-footprinted viruses.** A. AID favors cytidine deamination in a 5' WRC context. The WRC trinucleotide motif is depicted in three possible codon contexts on both coding and template strand. Depending on the position of the mutated C, the C to T transition can be synonymous (S) or non-synonymous (NS). Proportion of S and NS mutations is reported when the two types of mutation can be produced. B. The WRC and NNGYWN observed/expected ratios for 33,400 human viruses' genomes (from 870 unique species) were calculated, grouped by species and colored according to the Baltimore classification. Each point represents a unique viral genome. C. List of the putative AID-footprinted viral genes (displaying WRC or NNGYWN depletion) and belonging to an otherwise non-depleted viral genome.
(PDF)

**S11 Fig. Search for APOBEC1-footprinted viruses.** A. APOBEC1 favors cytidine deamination in a 5' WCW context. The WCW trinucleotide motif is depicted in three possible codon contexts on both coding and template strand. Depending on the position of the mutated C, the C to T transition can be synonymous (S) or non-synonymous (NS). Proportion of S and NS mutations is reported when the two types of mutation can be produced. B. The NWCWNN and NWGWNN observed/expected ratios for 33,400 human viruses' genomes (from 870 unique species) were calculated, grouped by species and colored according to the Baltimore

classification. Each point represents a unique viral genome. C. The observed/expected ratios of WCW trinucleotide at various codon positions and on both strands (i.e. NWCWNN, WCW, NNWCWN, NWGWNN, WGW and NNWGWN) were calculated for the NWCWNN and/or NWGWNN depleted viral species and depicted by a heatmap. A colored scale with increasing shades of blue indicating depletion and increasing shades of red indicating enrichment. P-values were calculated by Student's unpaired, two-tailed t-test (NS for not significant, * p< 0.05, ** p< 0.01, *** p< 0.001). D. List of the putative APOBEC1-footprinted viral genes (displaying NWCWNN or NWGWNN depletion) and belonging to an otherwise non-depleted viral genome.
(PDF)

**S12 Fig. A3 footprint in human oncogenic viruses.** NTC and NNGANN observed/expected ratios were calculated for each available coding sequence of eleven well-known cancer-related viruses. Each point represents a unique viral coding sequence. The coding sequences are grouped and colored according to gene name.
(PDF)

**S1 Table. Genomic K-mer ratios for human viruses.** Observed/expected K-mer ratios for each genomic human viral sequence (available for download at https://doi.org/10.5061/dryad.n8pk0p2sd).
(TXT)

**S2 Table. Genic K-mer ratios for human viruses.** Observed/expected K-mer ratios for each genic human viral sequence (available for download at https://doi.org/10.5061/dryad.n8pk0p2sd).
(ZIP)

**S3 Table. Genomic K-mer ratios for non-human viruses.** Observed/expected K-mer ratios for each genomic and genic non-human viral sequence (available for download at https://doi.org/10.5061/dryad.n8pk0p2sd).
(TXT)

## Acknowledgments

We thank members of the URVI lab for valuable discussions.

## Author Contributions

**Conceptualization:** Florian Poulain, Noémie Lejeune, Kévin Willemart, Nicolas A. Gillet.

**Data curation:** Florian Poulain.

**Formal analysis:** Florian Poulain.

**Funding acquisition:** Nicolas A. Gillet.

**Investigation:** Florian Poulain.

**Methodology:** Florian Poulain, Nicolas A. Gillet.

**Project administration:** Nicolas A. Gillet.

**Resources:** Nicolas A. Gillet.

**Software:** Florian Poulain.

**Supervision:** Nicolas A. Gillet.

**Validation:** Florian Poulain.

**Visualization:** Florian Poulain.

**Writing – original draft:** Florian Poulain.

**Writing – review & editing:** Nicolas A. Gillet.

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
