## [Decision Letter · Decision Letter 0]

3 Apr 2020

Dear Dr gillet,

Thank you very much for submitting your manuscript "Footprint of the host restriction factors APOBEC3 on the genome of human viruses" for consideration at PLOS Pathogens. As with all papers reviewed by the journal, your manuscript was reviewed by members of the editorial board and by several independent reviewers. In light of the reviews (below this email), we would like to invite the resubmission of a significantly-revised version that takes into account the reviewers' comments.

We cannot make any decision about publication until we have seen the revised manuscript and your response to the reviewers' comments. Your revised manuscript is also likely to be sent to reviewers for further evaluation.

Sincerely,

Paul Francis Lambert

Associate Editor

PLOS Pathogens

Karl Münger

Section Editor

PLOS Pathogens

Kasturi Haldar

Editor-in-Chief

PLOS Pathogens

orcid.org/0000-0001-5065-158X

Michael Malim

Editor-in-Chief

PLOS Pathogens

orcid.org/0000-0002-7699-2064

Reviewer's Responses to Questions

**Part I - Summary**

Reviewer #1: The manuscript “Footprint of the host restriction factors APOBEC3 on the genome of human viruses" by Poulain et al. expands upon previous work on classifying the depletion of APOBEC3 target motifs in over 24,000 human viruses and some non-human primate, avian, and fish viruses. Using methods adapted and expanded from Warren et al., the authors first validate that their analysis supports significant APOBEC3-motif depletion in polyomaviruses and papillomaviruses. They then identify that more viruses are depleted for this motif generally across their genomes. Additionally, some viruses that would normally not be considered depleted have significant depletion in certain regions of their genomes consistent with viral genome replication or transcription, such as adenoviruses and herpesviruses. This study is thorough and is an important comparison of APOBEC depletion using the same analysis approach across many virus families. There are some major concerns that need to be addressed first before I would recommend this interesting manuscript for acceptance.

Reviewer #2: To my knowledge this is the first valiant attempt to comprehensively analyze the APOBEC mutation signature across “all” viruses. This pan-virome analysis confirmed the documented presence of an APOBEC footprint on many viruses, most notably polyoma viruses and papillomaviruses. The approach also shed light on some less characterized APOBEC substrates including erythroviruses and coronaviruses (though curiously not those that recently zoonosed including human SARS-CoV-2). A gene-centric modification of the approach also indicated some interesting correlations with gamma-herpesvirus and adenovirus DNA replication strategies. Overall, the authors estimate an APOBEC impact on 16% of “all” viruses (though this number may be debatable because not all RNA viruses seem to be analyzed here).

Reviewer #3: Poulain and colleagues use a computational approach to identify a decrease in TC dinucleotides in a variety of viral genomes. The authors suggest that the decrease in TC, specifically NTC, represents an APOBEC3 footprint in these viral genomes. This has the potential of being a really interesting story, but the authors need to address a couple of concerns before publication is warranted.

**Part II – Major Issues: Key Experiments Required for Acceptance**

Reviewer #1: The taxonomic labeling of the supplemental tables is incorrect. The genus column contains family and subfamily classifications. The family column contains genus and the species column contains both species, type and strain. It is unclear how these discrepancies may have affected data analysis and plotting.

Additionally, I found an error in the annotation of one virus in the supplemental excel file. There may be more that could minorly affect the results:

ssRNA viruses Flaviviridae Alphapapillomavirus Tick-borne encephalitis virus

The authors should add more detail in the methods as to how the viral sequences were selected. Were identical sequences collapsed?

I’m not fully convinced that the Student’s t-test of NTC/NNGANN versus TCN/NGA & NNTCNN/GAN is a necessary comparison. What is the biological take home message from these comparisons?

Reviewer #2: Major

1) At multiple points the authors claim to have analyzed “all” human viruses. This is obviously a moving target with many new viruses being reported each year (ex. thousands in one paper alone from a recent paper by the Buck lab – PMID 32014111 – which could alone change the results of the present study). Therefore, I suggest tempering this claim to “all currently annotated viruses” or all “NCBI references viruses” (and state the download window or dates in the methods).

2) Lines 190-92 indicate no depletion of A3G preferred NCC sites in viruses. A close look at Fig 3b shows some viruses with a clear underrepresentation of NCC sites. Are these retros? The authors should also comment on outliers that maybe impacted by A3G. In other words, the wholistic view is appreciated but a closer look at exceptions would also help balance the story.

3) Authors should extend analyses of gene-level footprints to include A3G-preferred motifs (in the very least as an additional negative control).

4) To be as comprehensive as possible, the authors should also consider including the footprint of AID in this analysis, as the only family member preferring purine nucleobases on the 5’ side of the target cytosine (RC).

5) HIV-1, HIV-2 and SIV should be have a dedicated results subsection and should be included in intraviral comparisons such as those in fig S6. The APOBECs were discovered as retrovirus restriction factors and they continue to have an impact on these viruses. The paper cites contrasting publications on the APOBEC footprint in retroviruses and I’m curious what the results are using the methodology described here in this ms.

6) A supplementary table of all downloaded file accession numbers should be included to enable full reproducibility.

Reviewer #3: 1. My main concern is related to how the ‘synthetic genomes’ are constructed by concatenating the different coding genes. It is not clear how overlapping genes are handled. It is clear that selection will work differently when 2 genes overlap. E.g. TCN in frame 1 = NTC in frame 2. However, a change in frame 2, will have a huge impact in frame 1 and will likely not be tolerated. The authors need to take this into account in their analysis. This is particularly important for the gene level analysis presented in figures 7, 8, 9 and 10. For example, is HBZ really affected by APOBEC, or is the apparent los of TCs due to being the sole gene on the opposite strand, while being in an intron for 3 other genes?

2. I would like the authors to include a control for non-coding sequences. If the loss of NTC is truly correlated with the degenerate DNA code, one would expect that all TC, regardless of codon position, will be affected.

3. APOBEC3 typically converts TC to TTT. The authors do not provide evidence that TC reduction is accompanied by an increase in TT. I would like to see this data. Specifically, in the case of the Coronaviridae where the evidence in favor of APOBEC mutations is, in my opinion, minimal (see next comment).

4. The data in figure 4 is used to argue that in addition to dsDNA viruses, members of the Coronaviridae have reduced NTC content. The authors use two standard deviations around the mean of a bimodal distribution. The logic behind this choice is not completely obvious to me. It is clear that the average of this bimodal distribution is heavily skewed by a large group of viruses that are not TC depleted. However, there is minimal evidence that RNA is a substrate for APOBEC3. Indeed, there is more convincing evidence that APOBEC does not edit RNA. Therefore, the distribution is artificially skewed towards 0, which affects the validity of the 2xSD approach. I would like the authors to use a different approach to show that the TC depletion of HKU is real. Moreover, a recent paper showing that APOBEC3 restricst coronavirus replication, also showed no evidence of hypermutation upon overexpression of APOBEC, but points towards a non-canonical APOBEC function (1).

5. The authors suggest a difference between human CoV and zoonotic CoVs. The authors need to take the evolutionary history of these viruses into account. Furthermore, bats have a significantly expanded APOBEC repertoire compared to humans, so the authors need to explain why bat APOBEC would not edit the CoV genomes, but human APOBECs do.

(1) 1) Milewska, A., Kindler, E., Vkovski, P. et al. APOBEC3-mediated restriction of RNA virus replication. Sci Rep 8, 5960 (2018). https://doi.org/10.1038/s41598-018-24448-2

**Part III – Minor Issues: Editorial and Data Presentation Modifications**

Reviewer #1: Line 100: this should be tempered to be representatives from all known human virus families. The authors do not have all papillomavirus types (400+) and therefore cannot claim all viruses.

Considering the previous point, the claim that 16% of all viruses being A3-motif depleted may be incorrect. How was this calculated initially?

Figure 9. Is RPMS1 completely depleted or is this an error? The other ORFs that are labeled black in panel A should be a color that is not on the k-mer ratio color scale.

Lines 370-378: Non-human coronaviruses do not show a TC depletion signature, but do they show evidence for deamination by bat APOBECs? Bat retroviruses show more mutation for GG to AG rather than GA to AA according to Hayward et al. Mol. Biol. Evo. 2018. Unfortunately the RPo assay used in that paper doesn’t comprehensively test all C containing trinucleotides, so there is not a clear motif preference for each bat APOBEC3, but could be potentially inferred by expanding the motif analysis beyond TC.

In a similar vein to the previous comment, is it possible that APOBEC1 (target motif: YCY) is the primary DNA deaminase against viruses in the absence of APOBEC3 enzymes? What about AID (WRCN)?

Potentially out of the scope of this paper, but I am curious whether any mouse viruses show mouse APOBEC3 motif depletion (TYCN)?

The sentence on line 249 is unclear: “Nevertheless, the observation of a general depletion of the NGA codons precludes any conclusion regarding the presence of an A3 footprint on the negative strand (Fig 6)"

I completely agree that NTC/NNGANN is a benchmark for APOBEC3 activity on viruses over evolutionary time. Can the authors speculate on whether viruses with depletions in TC/GA motifs that would otherwise result in coding changes are more protected against APOBEC3 activity and therefore in less need of anti-APOBEC proteins such as Vif or BORF2.

Parvovirus B19 also perturbs the cell cycle and E2F family of genes through NS1 (PMID: 20890043, PMID: 282640280). For BKPyV (and likely relatives) this function seems to be essential for APOBEC3B upregulation. Does this differ from other parvoviruses as a potential explanation for the observed APOBEC3 motif depletion?

Do the codons in the Parvovirus B19 genome that overlap the 7.5k protein encode conserved amino acids for function or are these diverse?

Line 358: The sentence “This observation came as a half surprise." Is unnecessary.

References 34 and 38 out of order in the text.

428: edition should be editing

Reviewer #2: Minor

- The animal pictures are not necessary in Fig 6.

- Line 363 – “Parallelly” is not a word; try “In parallel” instead.

- Line 428 – “edition” is not a word; use “editing” instead.

- Nomenclature – use accepted names recent zoonotic virus - SARS-CoV-2 or COVID-19; not the much less common “Wuhan seafood market pneumonia virus”).

- The references are not yet in the correct format for PLoS Pathogens (and some like #5 and #11 and others are incomplete)

Reviewer #3: 1. The authors should consider moving Figure 1 and the associated paragraph moved to materials and methods.

2. Overall the figures are overly complicated. I do not feel like they really convey the results in an optimal manner. I would urge the authors to use less figures and improve their interpretability.

3. Figure 2, The authors conclude that both strands are edited. It is however clear that NTC is significantly less common than NNGANN. This should be addressed.

4. The authors compare human viruses to viruses in other groups of organisms. The authors should do this comparison for individual Baltimore classes. However, given the data in figure 4, it may not be relevant to compare anything but dsDNA viruses. Figure 3 should include some sort of statistical support.

PLOS authors have the option to publish the peer review history of their article (what does this mean?). If published, this will include your full peer review and any attached files.

Reviewer #1: No

Reviewer #2: No

Reviewer #3: No
---

## [Decision Letter · Decision Letter 1]

19 Jun 2020

Dear Dr gillet,

We are pleased to inform you that your manuscript 'Footprint of the host restriction factors APOBEC3 on the genome of human viruses' has been provisionally accepted for publication in PLOS Pathogens.

Best regards,

Paul Francis Lambert

Associate Editor

PLOS Pathogens

Karl Münger

Section Editor

PLOS Pathogens

Kasturi Haldar

Editor-in-Chief

PLOS Pathogens

orcid.org/0000-0001-5065-158X

Michael Malim

Editor-in-Chief

PLOS Pathogens

orcid.org/0000-0002-7699-2064

Reviewer Comments (if any, and for reference):

Reviewer's Responses to Questions

**Part I - Summary**

Reviewer #1: The authors' revisions have adequately addressed all of my concerns.

Reviewer #2: This is a comprehensive analysis of the APOBEC footprint in all available viral sequences. The authors have addressed my concerns fully.

**Part II – Major Issues: Key Experiments Required for Acceptance**

Reviewer #1: None

Reviewer #2: None

**Part III – Minor Issues: Editorial and Data Presentation Modifications**

Reviewer #1: None

Reviewer #2: None

PLOS authors have the option to publish the peer review history of their article (what does this mean?). If published, this will include your full peer review and any attached files.

Reviewer #1: No

Reviewer #2: No

---

## [Editor Report · Acceptance letter]

27 Jul 2020

Dear Dr gillet,

We are delighted to inform you that your manuscript, "Footprint of the host restriction factors APOBEC3 on the genome of human viruses," has been formally accepted for publication in PLOS Pathogens.

Best regards,

Kasturi Haldar

Editor-in-Chief

PLOS Pathogens

orcid.org/0000-0001-5065-158X

Michael Malim

Editor-in-Chief

PLOS Pathogens

orcid.org/0000-0002-7699-2064